# Genetic determinants of fatigue up to 2 years after radiotherapy in prostate cancer patients

**A list of authors and their affiliations appears at the end of the paper**

Fatigue is a common symptom of cancer patients, which can impair quality of life even years after treatment. Little is known about genetic determinants of fatigue, especially in prostate cancer (PCa). This study aims to identify SNPs associated with long-term fatigue in a prospective cohort of PCa patients. A genome-wide association study was conducted in non-metastatic PCa patients treated with radiotherapy in 7 European countries and the USA. A total of 1,381 men recorded fatigue using the EORTC QLQ-C30 and 877 men additionally completed the Multidimensional Fatigue Inventory (MFI) up to two years post-radiotherapy. Clinically important fatigue is defined for the EORTC QLQ-C30 based on the proposed threshold as scores ≥39 and for the MFI as scores ≥75th percentile in the general German male population aged ≥60 years. Regression models adjusted for demographic, disease- and treatment-specific characteristics examine SNPs associated with clinically important fatigue. Differential gene expressions are explored using expression quantitative trait analysis. rs142212041 located in the *ACTR3/CBWD2* gene region is significantly associated ($P = 3 \times 10^{-8}$) with long-term physical fatigue in 643 men without physical fatigue pre-radiotherapy. Several potential risk loci ($P < 5 \times 10^{-6}$) are identified for distinct fatigue phenotypes. Gene expression differences are observed for *ACTR3* and *CBWD2*, although not significant after correction for multiple testing. The results emphasise the multidimensionality of fatigue and suggest a plausible biological mechanism in fatigue pathophysiology, previously discussed for myalgic encephalomyelitis/chronic fatigue syndrome, which might be a potential intervention target.

Cancer-related fatigue is a common symptom experienced by cancer patients and survivors and is defined as 'a distressing, persistent, subjective sense of physical, emotional, and/or cognitive tiredness or exhaustion related to cancer or cancer treatment that is not proportional to recent activity and interferes with usual functioning'[1,2]. Various biological mechanisms in the aetiology of fatigue have been hypothesised, including the involvement of cytokine and hypothalamic-pituitary-adrenal axis dysregulation[3]. Prostate cancer (PCa) is the most common cancer in men in most European countries

and North America[4]. A substantial proportion of PCa patients and survivors experience fatigue during treatment and years thereafter, with reported prevalences ranging from 13% to 82%[5]. Patients affected by fatigue exhibit a variety of manifestations, including lack of energy and impaired concentration[6]. While fatigue may occur as a transient adverse event following treatment with radiotherapy and hormone therapy, some patients experience post-treatment fatigue for several years, which can substantially impair their health-related quality of life (HRQoL)[7,8]. Genome-wide approaches are increasingly used to

✉ e-mail: p.seibold@dkfz-heidelberg.de

**Table 1 | Selected characteristics of prostate cancer patients with available genetic data from the REQUITE cohort that assessed the EORTC QLQ-C30 Fatigue Scale and at least one Multidimensional Fatigue Inventory dimension in a sub-cohort up to two years following external beam radiotherapy**

| EORTC QLQ-C30 Fatigue Scale | | Multidimensional Fatigue Inventory |
| --- | --- | --- |
| Characteristic | N = 1381[a] | N = 877[a] |
| **Country** | | |
| Belgium | 296 (21%) | 0 (0%) |
| France | 211 (15%) | 209 (24%) |
| Germany | 71 (5%) | 70 (8%) |
| Italy | 150 (11%) | 149 (17%) |
| Netherlands | 61 (4%) | 0 (0%) |
| Spain | 241 (17%) | 242 (28%) |
| UK | 345 (25%) | 201 (23%) |
| USA | 6 (<1%) | 6 (1%) |
| **Age at enrolment (years)** | | |
| Median (Range) | 70 (46–88) | 70 (46–88) |
| **BMI (kg/m²)** | | |
| <25 | 356 (26%) | 213 (24%) |
| 25–30 | 719 (52%) | 467 (53%) |
| >30 | 301 (22%) | 196 (22%) |
| **History of depression[b]** | | |
| Yes | 106 (8%) | 49 (6%) |
| No | 1275 (92%) | 828 (94%) |
| **Smoking Status** | | |
| Never | 522 (38%) | 351 (40%) |
| Former | 690 (50%) | 418 (48%) |
| Current | 163 (12%) | 108 (12%) |
| **Tumour Size[c]** | | |
| T1/T2/MRI not visible | 356 / 656 / 4 (74%) | 278 / 424 / 4 (81%) |
| T3/T4 | 325 / 24 (25%) | 158 / 6 (19%) |
| **Nodal Involvement** | | |
| NX | 219 (16%) | 185 (21%) |
| N0 | 1095 (79%) | 653 (74%) |
| N1 | 65 (5%) | 39 (4%) |
| **Prostate-Specific Antigen (ng/mL), pre-diagnostic biopsy** | | |
| ≤10 | 757 (59%) | 512 (62%) |
| 11-20 | 293 (23%) | 183 (22%) |
| ≥21 | 227 (18%) | 137 (16%) |
| **ISUP Grade** | | |
| 1 (Gleason score: 2–6) | 234 (17%) | 168 (19%) |
| 2 (Gleason score: 3 + 4) | 454 (33%) | 286 (33%) |
| 3 (Gleason score: 4 + 3) | 339 (25%) | 235 (27%) |
| 4 (Gleason score: 8) | 171 (12%) | 93 (11%) |
| 5 (Gleason score: 9–10) | 178 (13%) | 93 (11%) |
| **D'Amico Risk Classification[d]** | | |
| Low Risk | 102 (8%) | 82 (10%) |
| Intermediate Risk | 601 (44%) | 420 (49%) |
| High Risk | 651 (48%) | 357 (42%) |
| **Prior Prostatectomy** | | |
| Yes | 420 (30%) | 236 (27%) |
| No | 961 (70%) | 641 (73%) |
| **Hormone Therapy** | | |
| None | 395 (29%) | 281 (32%) |
| Short-term (≤ 6 months) | 782 (57%) | 477 (54%) |
| Long-term (> 6 months) | 204 (15%) | 119 (14%) |
| **Radiotherapy** | | |
| EBRT | 1381 (100%) | 877 (100%) |

**Table 1 (continued) | Selected characteristics of prostate cancer patients with available genetic data from the REQUITE cohort that assessed the EORTC QLQ-C30 Fatigue Scale and at least one Multidimensional Fatigue Inventory dimension in a sub-cohort up to two years following external beam radiotherapy**

| EORTC QLQ-C30 Fatigue Scale | | Multidimensional Fatigue Inventory |
| --- | --- | --- |
| Characteristic | N = 1381[a] | N = 877[a] |
| **Dose per fraction (Gy)** | | |
| ≤2 | 1034 (75%) | 871 (99%) |
| >2 | 347 (25%) | 6 (1%) |
| **Total dose (Gy)** | | |
| ≤72 | 590 (43%) | 214 (24%) |
| 74–76 | 557 (40%) | 477 (54%) |
| >76 | 232 (17%) | 185 (21%) |
| **Pelvic Radiotherapy** | | |
| Yes | 434 (31%) | 333 (38%) |
| No | 947 (69%) | 544 (62%) |

BMI Body mass index; EBRT External beam radiotherapy; Gy Grey
[a] n(%);
[b] History of depression was defined as self-reported prior diagnosis and/or use of anti-depressants.;
[c] If available, clinical T stage was considered, else pathological T stage or MRI T stage.;
[d] The D'Amico risk classification classifies prostate cancer patients into low risk (clinical T stage T1c, T2a and PSA level ≤ 10 ng/mL and Gleason score of ≤ 6), intermediate risk (clinical T stage T2b or PSA level 11–20 ng/mL or Gleason score of 7) and high risk (clinical T stage T2c or PSA level > 20 ng/mL or Gleason score of ≥ 8) for biochemical recurrence following surgery[74].

investigate inter-patient variability in radiotherapy toxicity. Previous studies identified several single nucleotide polymorphisms (SNPs) associated with radiation toxicity endpoints in breast and prostate cancer patients[9–12]. While demographic and treatment-related determinants of long-term fatigue, such as younger age and hormone therapy, have been reported, little is known about genetic determinants of long-term fatigue in PCa[7,13].

The aim of this genome-wide association study (GWAS) was to identify SNPs associated with long-term fatigue, including distinct fatigue dimensions, up to two years after radiotherapy in a prospective, international cohort of non-metastatic PCa patients. SNPs associated with incident long-term fatigue were additionally investigated in the subgroup of men without fatigue before the start of radiotherapy. In an expression quantitative trait loci (eQTL) analysis, expression differences of genes in the identified risk loci between SNP-based genotypes were explored.

## Results
### Study population
1578 PCa patients enrolled in the REQUITE cohort prior to radiotherapy were treated with external beam radiotherapy (EBRT) without brachytherapy (Supplementary Fig. 1). Genetic data were available for 1508 patients. Data on the fatigue scale from the EORTC QLQ-C30 questionnaire were available for 1381 (1381/1508; 92%) participants with available genetic data, with 877 (877/1,508; 58%) participants reporting at least one of the five fatigue dimensions in the Multidimensional Fatigue Inventory (MFI) questionnaire at one and/or two years after the end of radiotherapy (Table 1). The median age at recruitment was 70 years (range: 46–88). 8% of men completing the EORTC QLQ-C30 and 6% completing the MFI questionnaires reported previous and/or current depression. The proportion of depressed men was higher among those fatigued prior to radiotherapy (Supplementary Data 1). Most patients had localised disease, no nodal involvement, and were in the intermediate- or high-risk groups for biochemical recurrence according to the D'Amico classification. Around two-thirds of men were treated with hormone therapy (ADT), with the majority

receiving short-term therapy. About one-third of men had previously undergone a prostatectomy.

20% (283) of 1381 PCa patients who completed the EORTC QLQ-C30 reported being fatigued one and/or two years following radiotherapy. 31% (274/876) of men with MFI assessments experienced physical, 31% (269/877) general, and 29% (256/873) mental fatigue. 28% reported reduced activity (247/874) and motivation (246/874). Among men without fatigue prior to radiotherapy, 15% (174/1147) reported being fatigued up to two years later with the EORTC QLQ-C30. Again, the proportions of affected men were greater with the MFI, with 22% (151/677) for general fatigue, 21% for physical fatigue (135/643), 20% for reduced motivation (138/676), and 19% for mental fatigue (129/666) and reduced activity (124/663).

## Association analysis

Of 21,465,139 genetic variants, 8,387,488 with an imputation score <0.3 and 5,981,339 with a minor allele frequency (MAF) < 0.05 were removed, leaving 7,096,312 common variants included in the analysis. Figure 1 shows the Manhattan plots of the fatigue phenotypes investigated. The associated QQ plots did not indicate genomic inflation ($\lambda \leq 1.04$) and indicated more variants associated with fatigue phenotypes than expected by chance (Supplementary Fig. 2). In the primary analysis, no genome-wide statistically significant associations were observed between the included SNPs and each of the fatigue phenotypes (Supplementary Data 2). Associations with $P < 5 \times 10^{-6}$ were found for the index SNPs rs74800609 ($P = 3.3 \times 10^{-7}$), rs17710795 ($P = 4.5 \times 10^{-7}$), and rs35612476 ($P = 9.5 \times 10^{-7}$) for general fatigue, rs1480885 ($P = 3.7 \times 10^{-7}$) for reduced activity, and rs35837098 ($P = 2 \times 10^{-7}$) for fatigue based on the EORTC QLQ-C30. The Manhattan plots further indicate several SNPs aligned with the index rs74800609 on chromosome 14 for general fatigue and index rs1480885 on chromosome 8 for reduced activity.

In men without clinically important pre-radiotherapy fatigue, SNP rs142212041 on chromosome 2 was statistically significantly associated with physical fatigue up to two years following radiotherapy (Fig. 2; Supplementary Data 3). For this SNP with a MAF of 17%, an odds ratio of 2.99 (95% confidence interval: 2.03 to 4.41; $P = 3.00 \times 10^{-8}$) was estimated. The Manhattan plot indicates two further potential loci with index SNPs rs7066898 ($P = 3.10 \times 10^{-7}$) and rs6553930 ($P = 8.40 \times 10^{-7}$), which did not reach genome-wide statistical significance. For the other fatigue phenotypes, multiple SNPs were found associated at $P < 5 \times 10^{-6}$, including rs12140921 ($P = 6.60 \times 10^{-7}$) for incident long-term reduced motivation, with several aligned SNPs in this region. The QQ plots with $\lambda \leq 1.04$ did not indicate genomic inflation (Supplementary Fig. 3).

## Internal validation and fine-mapping

The effect estimates for rs142212041 remained consistent in both direction and magnitude across subgroup and sensitivity analyses (Supplementary Table 1). Limiting the analysis to men with complete MFI assessments, a genome-wide statistically significant association with incident physical fatigue was also observed for rs7066898 (OR: 1.79; 95% confidence interval: 1.45 to 2.21; $P = 5 \times 10^{-8}$; Supplementary Fig. 4). The bootstrap odds ratio of 1.86 for the index SNP rs142212041 (standard error: 0.20; 95% confidence interval: 1.01 to 3.37) is consistent in direction with the non-bootstrap estimate, albeit of lower magnitude (Supplementary Fig. 5). The effect estimates of the leave-one-out analyses suggest that no country exclusively drives the signal for the index SNP (Supplementary Table 2). The LocusZoom plot shows log-transformed $P$-values from the association analysis for SNPs in the interval spanning 114.20 Mb to 115.00 Mb on chromosome 2 for the overall cohort and the subgroup of men without clinically important physical fatigue prior to radiotherapy (Fig. 3). The genome-wide statistically significant index SNP rs142212041 identified in men without physical fatigue before radiotherapy initiation is in the intronic region of the *ACTR3* Antisense RNA1 (*ACTR3-AS1*) gene. The variant is in

linkage disequilibrium (LD) with a cluster of SNPs in the chromosomal segment. The index SNP rs13394124 in the overall cohort is in LD with several SNPs in the chromosomal region, although not statistically significant.

The conditional analysis did not indicate secondary signals in the genomic region around the index SNP rs142212041. Fine-mapping of the susceptibility locus on chromosome 2 from 114.40 Mb to 114.80 Mb revealed two credible sets containing three putative causal variants: rs35060225, rs34241561 and rs12472252 (Supplementary Fig. 6).

## In Silico analyses

Local genetic correlations between incident physical fatigue and EHR-derived malaise and fatigue and chronic fatigue syndrome (ME/CFS) were examined. Non-associated loci on chromosome 2 were filtered using univariate analysis and a Bonferroni-corrected significance level threshold of $\alpha = 0.05/220 = 0.00023$. Bivariate tests were performed for the remaining loci. For fatigue and malaise (3413 cases, 403,789 controls), bivariate tests were calculated for 158 loci on chromosome 2 with a mean $r_g$ of 0.03. 19 loci showed a Bonferroni-corrected significant bivariate $r_g$, including the locus around the index SNP rs142212041 ($r_g = -0.09$, $P = 0.0001$). For ME/CFS (590 cases, 403,791 controls), bivariate tests were calculated for 158 loci, 18 of which showed a Bonferroni-corrected significant bivariate local genetic correlation. Across loci on chromosome 2, mean $r_g$ was 0.03. For the locus harbouring the index SNP, a highly statistically significant local genetic correlation between incident physical fatigue and ME/CFS was observed ($r_g = 0.18$, $P = 5.79 \times 10^{-13}$). In the locus 200 kb around *ACTR3*, the $P$-values of the genome-wide association analysis and cis-eQTLs correlated most strongly for skeletal muscle tissue (r = 0.79, P = 0.02; Supplementary Fig. 7) and whole blood (r = 0.63, P = 0.10; Supplementary Fig. 8). No statistically significant associations were observed for pituitary gland (r = 0.47, P = 0.21; Supplementary Fig. 9) and hippocampal tissue (r = −0.17, P = 0.89; Supplementary Fig. 10). Across tissues, no evidence was found that the trait-significant variant rs142212041 was associated with *ACTR3* expression levels.

SNP annotation was performed for the index SNP rs142212041, candidate causal variants rs35060225, rs34241561, rs12472252, and variants with $r^2 \geq 0.8$. Annotated variants showed increased H3K27ac enhancer activity in astrocytes (rs35060225, rs34241561) and brain tissues, including anterior caudate (rs10210927, rs12475444), hippocampus (rs12475444) and substantia nigra (rs12475444).

In the gene-based association analysis, no genes reached statistical significance with an false discovery rate (FDR)-adjusted $P < 0.05$. The most significant genes (unadjusted $P < 0.001$) are shown in Supplementary Fig. 11. Functional enrichment of putative genes mapped from top GWAS SNPs suggested involvement of processes related to inflammation and cytoskeletal dynamics, including lamellipodium assembly/organisation (Supplementary Fig. 12, Supplementary data 4).

## Heritability analysis

In an exploratory genome-wide complex trait analysis, the highest heritability estimate was observed for the EORTC QLQ-C30 fatigue scale ($H^2$: 0.31, SE: 0.18), which was statistically significant ($P = 0.04$), followed by reduced activity ($H^2$: 0.22, SE: 0.27, $P = 0.19$), mental ($H^2$: 0.20, SE: 0.29, $P = 0.25$) and physical ($H^2$: 0.17, SE: 0.28, $P = 0.27$) fatigue. The lowest estimates were obtained for general fatigue ($H^2$: <0.01, SE: 0.27, $P = 0.50$) and reduced motivation ($H^2$: <0.01, SE: 0.28, $P = 0.50$).

## Differential gene expression analysis

Differences in gene expression based on SNP-based genotypes were further explored in a different subset of the REQUITE PCa population. eQTL analysis was applied to 12 genes of 27 genes spanning the risk locus from 114.20 Mb to 115.00 Mb on chromosome 2, after excluding those with zero counts. After hard-calling the genotypes, 70 (58%) of 121 PCa patients with available dosage and expression data were

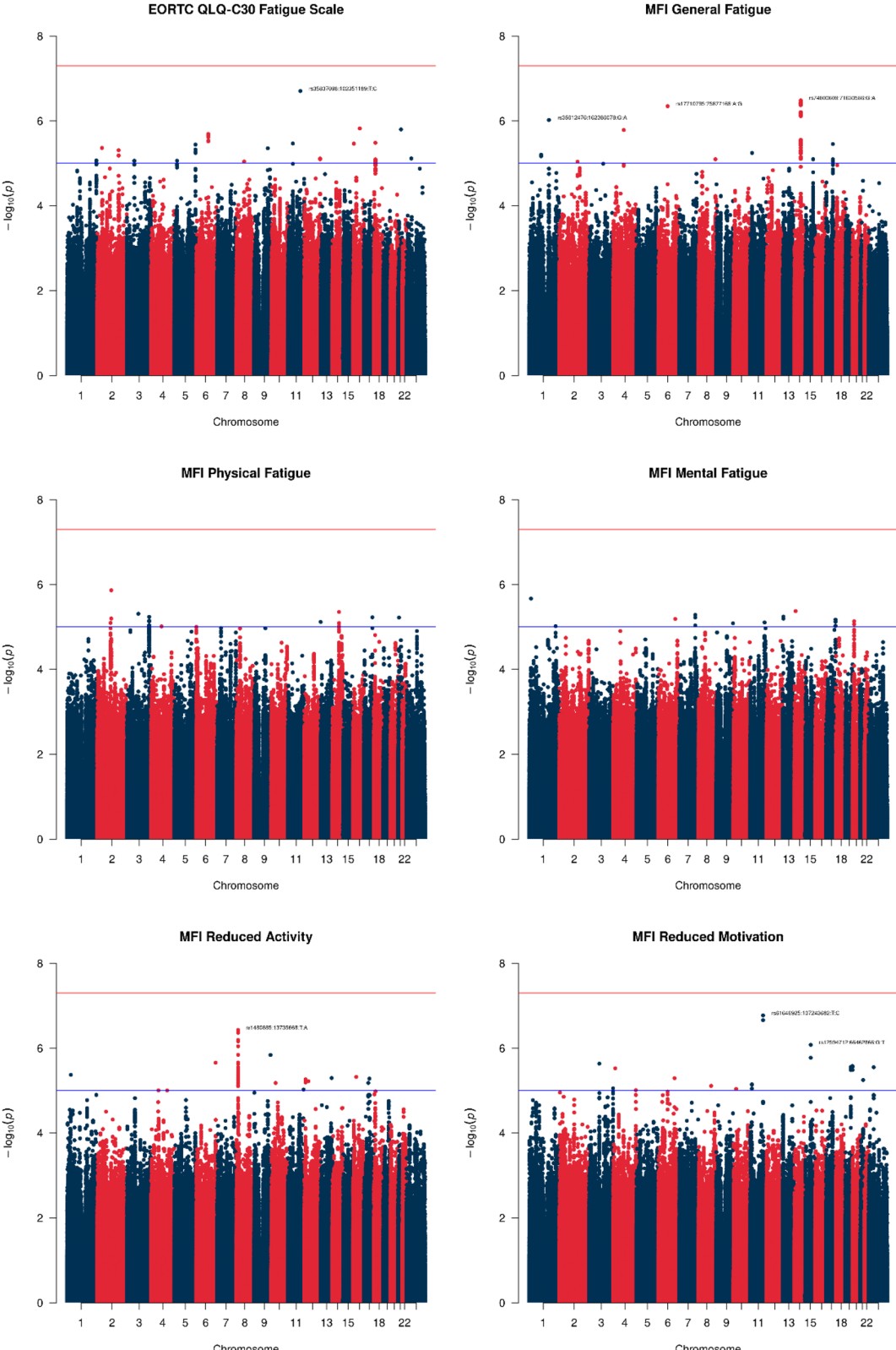

**Fig. 1 | Manhattan plots for fatigue in the EORTC QLQ-C30 (_N_ = 1284) and distinct fatigue dimensions in the Multidimensional Fatigue Inventory (_N_ = 826) up to two years following external beam radiotherapy in prostate cancer patients of the REQUITE cohort.** Multivariable regression models were adjusted for age and BMI at enrolment, baseline fatigue level prior to radiotherapy, depression, hormone therapy, prostatectomy, tumour size, nodal involvement, pelvic radiotherapy, and the top 5 principal components. The red line indicates genome-wide statistical significance ($P < 5 \times 10^{-8}$) and the blue line indicates suggestive significance ($P < 5 \times 10^{-6}$). Source data are provided as a Source Data file.

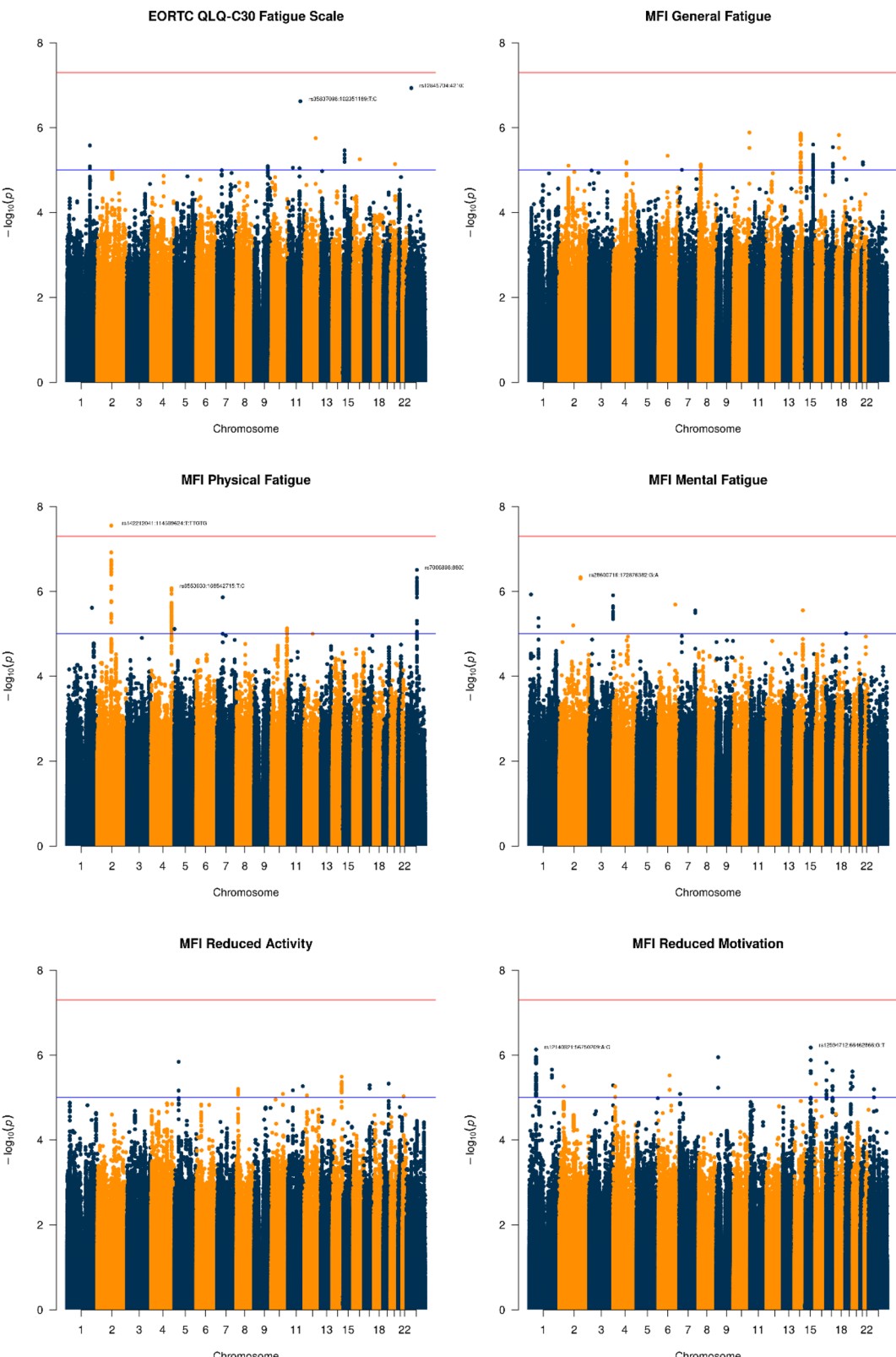

**Fig. 2 | Manhattan plots for fatigue in the EORTC QLQ-C30 (N = 1147) and distinct fatigue dimensions in the Multidimensional Fatigue Inventory (up to N = 677) up to two years following external beam radiotherapy in prostate cancer patients of the REQUITE cohort without fatigue before the start of radiotherapy.** Multivariable regression models were adjusted for age and BMI at enrolment, depression, hormone therapy, prostatectomy, tumour size, nodal involvement, pelvic radiotherapy, and the top 5 principal components. The red line indicates genome-wide statistical significance (P < 5 × 10⁻⁸) and the blue line indicates suggestive significance (P < 5 × 10⁻⁶). Source data are provided as a Source Data file.

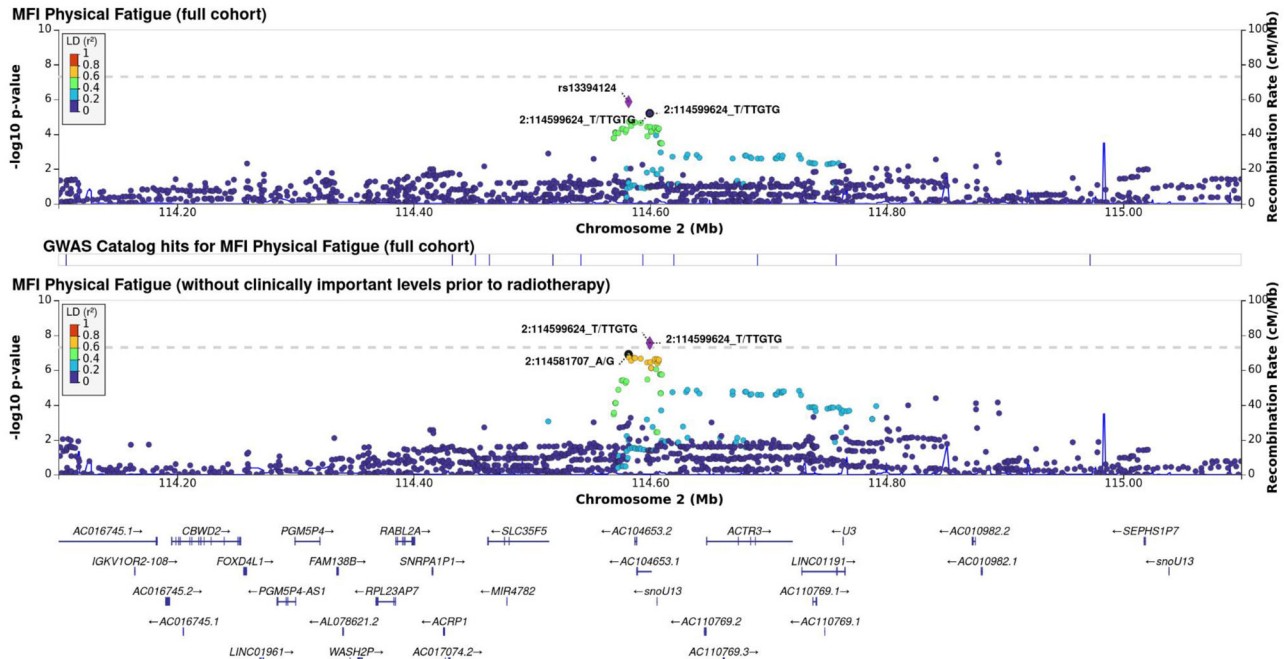

**Fig. 3 | LocusZoom plot with log-transformed P values of the association analyses, linkage disequilibrium, and recombination rate for the identified risk locus on chromosome 2 associated with incident long-term physical fatigue for the overall cohort of prostate cancer patients and for the subgroup of prostate cancer patients without physical fatigue before radiotherapy in the REQUITE cohort.** The respective index SNP is depicted as a purple diamond. The dotted line indicates genome-wide statistical significance ($P < 5 \times 10^{-8}$). Source data are provided as a Source Data file.

included in the analysis. 60 patients (86%) were of homozygous (A/A) and 10 patients (14%) of heterozygous (A/B) genotype. Differential expression patterns were observed for the genes *CBWD2* and *ACTR3* with log2 fold changes of −0.17 and −0.09 ($P < 0.04$), respectively, which were not statistically significant after correction for multiple testing (Supplementary Table 3; Fig. 4).

## Discussion

The aim of this study was to identify SNPs associated with fatigue up to two years after radiotherapy in men with non-metastatic PCa. Several potential risk loci across the different fatigue dimensions in the EORTC QLQ-C30 and MFI were identified, reflecting the complexity of fatigue as a polygenic trait that manifests as a variety of clinical symptoms. A SNP in the intronic region of the *ACTR3-AS1* gene reached genome-wide statistical significance and was associated with incident long-term physical fatigue. The *ACTR3* gene encodes the actin-related protein 3, which is a component of the ARP2/3 complex[14]. Actin-related proteins play a critical role in processes including vesicle motility and modulation of chromatin structure[15]. Previously, long non-coding RNAs associated with fatigue in PCa were reported, which were involved in actin filament and chromatin organisation[16]. While little is known about the *ACTR3* antisense RNA, the *ACTR3* gene was shown to be differentially expressed in women with major depressive disorder and patients with ME/CFS[17,18]. Furthermore, ME/CFS patients showed an increased number of extracellular vesicles (EVs)—lipid membrane vesicles without a functional nucleus released by cells—with significantly upregulated actin network proteins compared to patients with idiopathic chronic fatigue or depression[19–21]. In a recent study, changes in EV cytoskeletal proteins correlated with symptom severity in female ME/CFS patients. Moreover, post-exercise changes in proteins related to actin polymerisation, including ACTR3, were negatively associated with arthralgia[22]. Therefore, a disruption of actin signalling in EVs could play a role in ME/CFS pathophysiology and warrant further investigation in relation to fatigue, where previously discussed mechanisms include disruption of the hypothalamic-pituitary-adrenal axis and

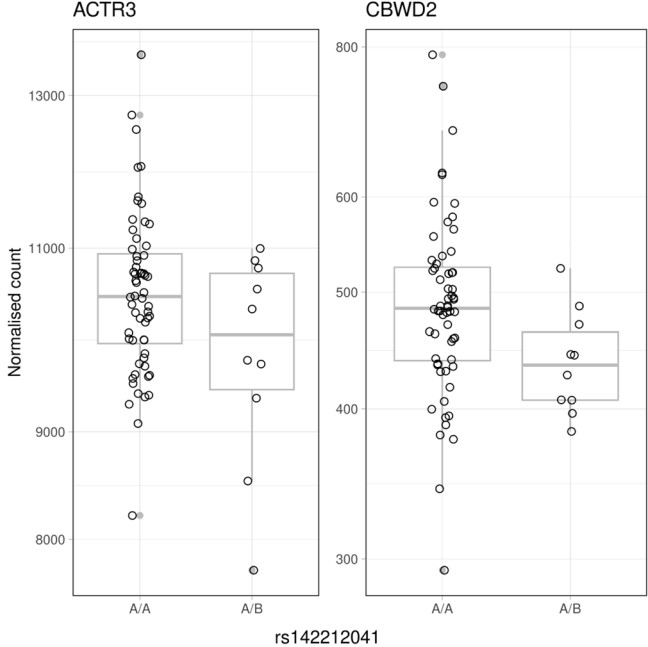

**Fig. 4 | Counts normalised by estimated size factors, including boxplots, from 70 non-metastatic prostate cancer patients in the REQUITE cohort stratified by SNP-based genotypes associated with incident long-term physical fatigue.** The median is shown as a horizonal line, with the bounds of the box depicting the 25th and 75th percentiles and the whiskers depicting the minimum and maximum values. Source data are provided as a Source Data file.

dysregulation of cytokines and serotonin[19,22,23]. Consistently, a highly statistically significant local genetic correlation between incident physical fatigue and ME/CFS was observed for the locus harbouring the index SNP rs142212041, albeit of weak magnitude. Functional studies

could prioritise this locus to investigate shared mechanisms underlying the aetiology of both phenotypes. GWAS and GTEx cis-eQTL signals correlated strongly in skeletal muscle tissue, although the trait-significant variant was not associated with *ACTR3* expression levels. However, GTEx samples were collected from deceased subjects, which may affect expression data[24]. In patients of the REQUITE cohort treated with radiotherapy, differences in the expression of the *ACTR3* and *CBWD2* genes were observed between SNP-based genotypes in whole blood, although not statistically significant after correction for multiple testing. Due to the hard-calling of imputed dosages, only a subset of samples (58%) with expression data were included. This might have limited the detection of subtle expression differences. Yet, the results may inform further research on fatigue and the genes in the identified risk locus, which is in proximity to a region between the *PAX8* and *CBWD2* genes, which has been previously shown to be linked to usual sleep duration[25]. Consistent with recent findings, functional analysis of regulatory elements indicated increased enhancer activity in astrocytes and several brain regions[26]. The gene-based analysis did not reveal significant associations; however, functional enrichment analysis of putative genes mapped from top GWAS SNPs suggested Gene Ontology processes related to lamellipodium assembly/organisation. Lamellipodia are flat, actin-driven membrane protrusions that guide cell movement and support processes such as satellite cell migration during muscle repair[27]. Their dynamics may also be relevant in fatigue states.

Heritability estimates suggest that genetic factors may contribute to fatigue in PCa patients. The proportion of phenotypic variation explained by common SNPs varied between fatigue dimensions and was generally higher than estimated for non-cancer populations[28]. Large standard errors, which may be inflated due to the small sample size, limit the conclusions that can be drawn and emphasise the need for collecting fatigue assessments in larger cohorts.

While previous studies have investigated genetic determinants of fatigue in other cancers, this is, to the best of the authors' knowledge, the first genome-wide association study to investigate long-term fatigue in a PCa patient population[29,30]. Previous genetic analyses in PCa mostly investigated fatigue at the end of treatment or shortly thereafter, and reported several associated genes. In non-metastatic PCa patients treated with EBRT, differential expression of the *SNCA* gene, associated with neuroinflammation, was linked to changes in fatigue symptoms[31]. Genes associated with cell damage pathways, including *BCL2L1*, *GADD45A* and *TGM2*, were also shown to be upregulated in fatigued men with PCa between the start and end of radiotherapy, although no statistically significant differences were observed compared to non-fatigued patients[32]. In predominantly metastatic PCa patients treated with ADT plus a variety of other treatments, including chemotherapy and palliative radiotherapy, 46 genes representing pathways including adrenergic, monoamine and peptides; immune response and inflammation; metabolites of muscle activity; lipid/energy metabolism, as well as transcription and growth factors were investigated. Compared to ME/CFS patients and non-fatigued controls, differences in gene expression were observed for 10 genes across each pathway studied[33].

The presented work has some limitations. The dichotomisation of fatigue assessments and the sample size, modest for a genome-wide analysis, limited the statistical power to identify genetic variants with smaller effect sizes. Furthermore, low-frequency and rare variants with MAF < 0.05 were not included in this analysis. The available data do not allow for establishing radiotherapy as causative for the observed fatigue, limiting the interpretation of the findings. However, the genome-wide statistically significant SNP for physical fatigue was observed in the subgroup of men without fatigue prior to radiotherapy after adjustment for demographic, disease- and treatment-specific characteristics. Therefore, it might be assumed that the fatigue that developed during radiotherapy is largely related to the treatment, although other factors such as anxiety, stress, pain, or insomnia due to the cancer diagnosis may have contributed. As no validated threshold for clinically important fatigue was previously published for the MFI, patients were classified as fatigued and non-fatigued using population-based reference values, as opposed to sample-based cut-offs. Measurement error, as well as comorbidities and lifestyle and psychosocial factors not fully captured, may lead to residual confounding. Although the enrichment analysis suggested plausible pathways potentially involved in fatigue, the results are strongly influenced by *ACTR3*. Among the strengths of the presented study are the comprehensive data collected in REQUITE, which allowed adjustment for a variety of potential confounding demographic, disease, and treatment factors. Due to the prospective fatigue assessment using the validated EORTC QLQ-C30 and MFI questionnaires, several fatigue phenotypes could be analysed, reflecting the complexity of the symptom.

Research is needed to confirm the findings in a further cohort of PCa patients treated with more recent therapies. In addition, potential differences in gene involvement in acute and long-term fatigue could be investigated. It is still unclear whether the findings can be generalised to patients with metastatic disease at diagnosis, other tumour entities, or populations of non-European ancestry. The identified susceptibility locus near *ACTR3/CBWD2* points to a plausible mechanism, which remains unconfirmed without functional corroboration. Successful replication of the identified fatigue susceptibility locus could inform the development of prediction models and polygenic risk scores, as well as facilitate the early identification of patients under elevated risk for long-term physical fatigue in clinical practice. Further analyses of the expression differences of genes in the identified chromosomal region may contribute to the understanding of fatigue pathophysiology, which could ultimately guide potential drug target discovery.

In conclusion, this study emphasises the multi-dimensionality of fatigue and has provided initial evidence for its heritability across different fatigue phenotypes. One SNP reached a genome-wide statistically significant level in association with incident physical fatigue up to two years following radiotherapy. This will require further validation in other PCa cohorts. In light of previous findings in ME/CFS and major depressive disorder, these results suggest a plausible biological mechanism in fatigue pathophysiology, which merits further investigation and might be a potential target for intervention.

## Methods
### Study population
Written informed consent was obtained from all patients. Local ethics committees approved the study, and it was registered under https://www.controlled-trials.com ISRCTN98496463.

Data from the prospective, international patient cohort REQUITE were analysed[34]. 1760 men with non-metastatic PCa were enrolled from 17 radiation oncology departments in seven European countries and the USA between 2014 and 2016 before the start of radiotherapy. Patients were treated with curative intent by EBRT and/or brachytherapy, with some patients receiving hormone therapy and/or prior prostatectomy. Demographic, lifestyle, cancer and treatment-related characteristics were prospectively collected. Patient-reported HRQoL was collected using the EORTC QLQ-C30 at baseline before radiotherapy initiation, at the end of radiotherapy, and annually thereafter until at least two years later[35]. In six out of eight countries, patient-reported fatigue was additionally assessed using the MFI at the same time points[36].

Participants with available genome-wide genotypes were included in the analysis if they received EBRT only (therefore, excluding those who received brachytherapy or a brachytherapy boost) and provided at least one EORTC QLQ-C30 fatigue scale and/or MFI dimension assessment one and/or two years after radiotherapy.

## Genotyping, imputation, and quality control

Patients were genotyped in one batch using the Illumina OncoArray-500K BeadChip with 600,000 SNPs. The array includes a genome-wide backbone of 300,000 SNPs and a comparable number of additional cancer-specific SNPs, of which 1000 SNPs were derived from previous radiogenomics studies in PCa. Genotype data were imputed according to the methods of the OncoArray network[37]. The 1000 Genomes Project (GP) Phase 3 (Haplotype release date October 2014) was used as the reference dataset for the autosomal chromosomes. The 1000 GP Phase 1 dataset (Haplotype ChrX release date August 2012) was used for the gonosomes[38]. Following a two-step imputation procedure using SHAPEIT and IMPUTE2, standard quality control procedures were performed[39,40]. Due to call rates <0.8 and <0.95, respectively, 9 and 13 samples and 7132 and 4874 variants were excluded from the REQUITE cohort. 1610 variants were excluded due to a MAF < 0.01 and a call rate < 0.98. 2746 variants were excluded due to deviations from the Hardy-Weinberg Equilibrium ($P < 10^{-7}$). 5 unexpected sample duplicates were excluded, as well as 7 samples with discrepancies between recorded and genomic sex. After a principal component analysis with EIGENSTRAT, 167 samples with <80% European ancestry were excluded[41]. To facilitate comparability of findings with previous studies on toxicities after radiotherapy, the analysis was based on common SNPs with MAF ≥ 0.05 and imputation score ≥ 0.3[42–44].

## Assessment of patient-reported fatigue

The EORTC QLQ-C30 questionnaire consists of multi-item symptom and functional scales, a global health status scale and individual symptoms commonly reported by cancer patients. Scores ranging from 0 to 100 were calculated for the fatigue scale according to the EORTC QLQ-C30 Scoring Manual based on 3 items, with a higher score indicating a higher fatigue level[45]. For the MFI, assessing multiple distinct dimensions of fatigue, scores ranging from 4 to 20 were calculated for the dimensions: general fatigue, physical fatigue, mental fatigue, reduced activity and reduced motivation. In both questionnaires, missing items were replaced by mean values of the completed items if at least half of the items of the scale were provided by a participant[45]. Prevalent fatigue in the EORTC QLQ-C30 was defined based on the proposed threshold for clinically important levels as scores ≥39 at one and/or two years after the end of radiotherapy[46]. As no validated threshold for clinically important levels is available for the MFI, prevalent fatigue was defined as scores ≥75th percentile at one and/or two years after the end of radiotherapy in the general German male population aged ≥60 years[47].

## Association analysis

Multivariable logistic regression models were used to investigate associations between SNPs and long-term fatigue up to two years following radiotherapy for the fatigue scale in the EORTC QLQ-C30 and the fatigue dimensions in the MFI. Covariates were selected based on potential clinical relevance. Models were adjusted for age and BMI at enrolment, baseline fatigue level prior to radiotherapy, depression, ADT, prostatectomy, tumour size, nodal involvement, pelvic radiotherapy, and the top 5 principal components to control for European population structure reflecting genetic ancestry. As secondary analysis, incident long-term fatigue was investigated in the subgroup of men without clinically important pre-radiotherapy fatigue levels. The logistic regression models were adjusted for the same covariates as in the primary analysis, excluding baseline fatigue level. As subgroup analyses, the genome-wide association analysis was stratified by receipt of ADT and pelvic radiotherapy and repeated separately for fatigue assessments at one year or at two years after radiotherapy. Additionally, the threshold for clinically important fatigue was raised from scores ≥75th percentile to scores ≥90th percentile in the general German male population aged ≥60 years to examine a more stringent threshold in fatigue classification. Non-responder bias was investigated by restricting the analysis to patients with complete fatigue

assessments. As sensitivity analyses, the association analysis was repeated with a reduced covariate set and extended by five additional genetic ancestry principal components.

Age and BMI at enrolment were included as continuous variables in the regression models. Clinically important baseline fatigue levels prior to radiotherapy, depression (self-reported prior diagnosis and/or use of antidepressants), prostatectomy, tumour size (MRI not visible/T1/T2 vs T3/T4; If available, clinical T stage was considered, else pathological T stage or MRI T stage.), nodal involvement (NX/N0 vs. N1), and pelvic radiotherapy (yes vs. no) were dichotomised in the models. ADT (none vs. short-term (≤ 6 months) vs. long-term (> 6 months)) was modelled ordinally. Patients with complete data were included in the regression models. $P$-values < 5×$10^{-8}$ were considered genome-wide statistically significant. PLINK 2.0 was used for the preparation of genetic data and regression modelling[48]. R 4.3.1 and the package qqman were used for creating Manhattan and QQ plots[49,50]. LD, recombination rate, and nearby genes of regions with genome-wide statistically significant SNPs were visualised using LocusZoom[51].

## Internal validation and fine-mapping

Bootstrap odds ratios and standard errors were calculated using a parametric bootstrap approach and 3000 replicates[52]. 95% confidence intervals were computed using conditional likelihood methods as implemented in the R package winnerscurse. In a leave-one-out analyses at the country level, the genome-wide association analysis was repeated by omitting one country respectively. Additional independent signals were examined in conditional analyses adjusted for genome-wide statistically significant SNPs. Fine-mapping was conducted using the Sum of Single Effects model and the package susieR[53].

## In Silico analyses

Shared genetic architecture between fatigue phenotypes in the cancer population studied, and EHR-derived malaise and fatigue and ME/CFS in white British participants of the UK Biobank was investigated using publicly available GWAS data[54,55]. Local genetic correlation, $r_g$, was calculated for susceptibility loci using the 1000 Genomes Project (EUR) Phase 3 as LD reference. Non-associated loci were filtered using univariate analysis and a Bonferroni-corrected significance level threshold, with bivariate tests performed for the remaining loci. The analysis was conducted using the package LAVA[56]. Colocalisation was explored using multi-tissue eQTL data derived from brain (hippocampus), pituitary gland, and skeletal muscle tissue and whole blood (Genotype-Tissue Expression (GTEx) project v7)[57]. The analysis was conducted using the R package eQTpLot[58]. Functional annotation was performed for regions harbouring genome-wide association signals, candidate causal variants, and variants with $r^2 ≥ 0.8$ using HaploReg v4.2 and Roadmap data[59,60].

Gene-based associations were assessed using MAGMA (v1.10) with GWAS summary statistics[61]. Genes were defined according to their genomic coordinates, including 35 kb upstream and 10 kb downstream regions to capture potential regulatory variants. SNP-level associations within each gene were aggregated using the mean SNP association model, which computes the average of SNP Z-scores while accounting for LD through the SNP covariance matrix. LD was estimated using a reference panel from European ancestry individuals from the 1000 Genomes Project Phase 3. In the analysis, genes with a FDR-adjusted $P < 0.05$ were considered statistically significant. The R package qqman was used for creating the Manhattan plot.

Functional enrichment analyses were conducted using two gene sets: i) candidate genes mapped from the locus in proximity to the top GWAS-associated SNP as annotated by dbSNP, and ii) genes identified through MAGMA analysis with nominal significance (unadjusted $P ≤ 0.05$). Over-representation testing was conducted using cluster-Profiler, ReactomePA, and related R packages across multiple

databases, including Gene Ontology (biological process, cellular component, molecular function), KEGG, Reactome, and curated MSigDB collections (BIOCARTA, CGP, ImmuneSigDB, transcription factor and microRNA targets, WikiPathways)[62–67]. Enrichment results were exported and visualised with custom R scripts.

## Heritability analysis

SNP heritability, as the proportion of phenotypic variation due to genetic variance between individuals, was quantified in a genome-wide complex trait analysis. Genetic relationship matrices were constructed using 6,883,788 SNPs on the autosome, from which heritability was estimated using a restricted maximum likelihood (REML) analysis. The analysis was adjusted for the top 5 principal components and conducted using the software package GCTA[68].

## Differential gene expression analysis

**Patients and samples.** REQUITE participants recruited in Manchester (United Kingdom) and Ghent (Belgium) were included. Whole blood RNA was extracted from PAXgene® Blood RNA Tubes using the PAXgene Blood miRNA Kit (*Qiagen*). Samples with RNA integrity RIN ≥ 6 and ~500 ng were considered. A total of 122 samples passed quality control and were included in the analysis.

**Libraries preparation and RNA seq.** To minimise technical artifacts, globins and ribosomal RNA were depleted using the *Ribo-Zero™ Gold Kit* (*Illumina*), and sequencing libraries were prepared with *TruSeq Stranded Total RNA kits* (*Illumina*). Paired-end (150 bp) sequencing was performed on the *NovaSeq6000* platform (*Illumina*), achieving an average depth of 60 million reads per sample. Sequence quality was assessed using *FASTQC* and *MULTIQC*, followed by alignment to the reference genome (GENCODE Release 38 [GRCh37]) using *STAR*[69,70]. Gene expression levels were normalised with R package *DESeq2* available via the Bioconductor repository, removing low-count genes, and batch effects were corrected using the RUVg method with empirical control genes[71].

**Gene expression analysis.** Gene expression was compared between genotypes by fatigue-associated SNPs identified in the GWAS. SNP-based genotypes were defined by hard-calling the imputed dosages using a probability threshold of 0.1[72]. Candidate genes were selected based on proximity to the risk loci identified in the GWAS. The differential gene expression analysis was performed using *DESeq2*, *P*-values were corrected for multiple testing using the Benjamini-Hochberg procedure ($q < 0.1$).

Results are reported following the STROGAR guideline[73].

## Reporting summary

Further information on research design is available in the Nature Portfolio Reporting Summary linked to this article.

# Data availability

Researchers may request access to the data analysed in this study for the purpose of cancer research. Access may be granted following a scientifically sound application and approval process managed by the REQUITE Publication Committee, which evaluates requests based on criteria such as alignment with patient consent, applicable data protection laws, and other relevant requirements. Fees may apply. Contact requite@leicester.ac.uk for more information. Source data are provided with this paper.

# Code availability

No custom codes were developed for this analysis; instead, publicly available software was used as referenced in the manuscript and the reporting summary.

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

## Acknowledgements

REQUITE received funding from the European Union's Seventh Framework Programme for research, technological development, and demonstration under grant agreement No. 601826. Funding for the gene expression analysis was enabled by ERAPerMed JTC2018 funding (ERAPERMED2018-244 and Spanish Instituto de Salud Carlos III (ISCIII-AC18/00117)). SG was supported by ERAPerMed JTC2018 funding (ERAPERMED2018-244 and SLT011/18/00005) and the Generalitat de Catalunya (2021SGR01112). AV research is supported by ISCIII funding, an initiative of the Spanish Ministry of Economy and Innovation partially supported by European Regional Development FEDER Funds (PI25/00744, PI22/00589, INT24/00023, DTS24/00083), the Spanish Association Against Cancer AECC (PRYES211091VEGA), and the Autonomous Government of Galicia (Consolidation and structuring programme: IN607B-2025-09). CW is supported by the UK Manchester NIHR Biomedical Research Centre and Cancer Research UK (C1094/A18504, C147/A25254). TRat and CJT are supported by the National Institute of Health Research (NIHR) Leicester Biomedical Research Centre. TRat was a previous Clinical Lecturer (CL-2017-11-02) and was also previously funded by a NIHR Doctoral Research Fellowship (DRF-2014-07-079). The views expressed are those of the authors and not necessarily those of the NIHR or the Department of Health and Social Care. The high-performance computing facilities at the University of Leicester (ALICE) were used to conduct this research. VHIO acknowledge the Cellex Foundation for providing research facilities, the CERCA Programme/Generalitat de Catalunya for institutional support, and the Agencia Estatal de Investigación for their financial support as a Centre of Excellence Severo Ochoa (CEX2020-001024-S/AEI/10.13039/501100011033). PS was supported by ERA-NET ERA PerMed 2018 funding JTC2018-244 (BMBF/01KU1912) and BfS funding (#3619S42261). We would like to thank all patients and patient representatives in the REQUITE study for their participation, as well as the reviewers for their thoughtful comments.

## Author contributions

Study planning: P.H., P.S., J.C.C., T.Rat., and C.J.T. Statistical evaluation: P.H., P.S., and M.E.A.B. Data analysis: P.H. and M.E.A.B. Data interpretation: P.H., P.S., M.E.A.B., J.C.C., R.K., S.L.K., M.E.S., A.W., T.R., T.Rat., A.V., and C.J.T. Writing of original draft: P.H., P.S., and M.E.A.B. Critical revision of the manuscript: H.K.J., E.B., R.B., J.C.C., A.M.D., O.F.R., S.G.E., S.L.K., J.C.R.R., M.E.S., E.S., R.P.S., L.V., M.R.V., T.W., A.W., C.M.L.W., T.R., T.Rat., A.V., and C.J.T. Patient enrolment and data collection: P.S., M.E.A.B., D.A., A.C., A.M.D., L.F., O.F.R., A.G.C., S.G.E., M.L., B.S.R., D.D.R., E.S., R.P.S., L.V., M.R.V., A.W., C.M.L.W., T.R., T.Rat., A.V., and C.J.T. Study and data management: P.S., J.C.C., A.W., and C.J.T. Patient advocates of the REQUITE study: E.B., H.S., and T.W.

## Funding

## Competing interests

AC: BMJ Editor in Chief of BMJ Oncology, 08/2022 to present – Employment. Grant/Research Funding from: Cancer Research UK Research, as PIE: Oct 2021 to March 2028; NIHR Research, as PIE: Dec 2023 to Sept 2030; Prostate Cancer UK Research, as PIE: May 2022 to April 2029; Elekta AB Research, as PIE: Jan 2019 to Jan 2029; Medical Research Council (MRC) Research, as PIE: 2018 ongoing; Astex Pharmaceuticals, Joint Lead applicant: Oct 2021 to Aug 2026; Yorkshire Cancer Research, YCR Research Award, Co-applicant: Aug 2022 to Nov 2030 Honoraria for single events around educational talks for commercial organisations: Astra Zeneca: 2023 October to 2025; Merck: 01/2023 - November 2025. DDR: No financial or non-financial competing interests as defined by Nature Portfolio or other interests that might be perceived to influence the interpretation of the article. AstraZeneca Research grant/ support/ Advisory Board: Institutional financial interests (no personal financial interests) BMS Research grant/ support/ Advisory Board: Institutional financial interests (no personal financial interests) Beigene Research grant/ support/ Advisory Board: Institutional financial interests (no personal financial interests) Philips Research grant/ support/ Advisory Board: Institutional financial interests (no personal financial interests) Olink Research grant/ support/ Advisory Board: Institutional financial interests (no personal financial interests) Eli-Lilly Advisory Board: Institutional financial interests (no personal financial interests) Pfizer Support: Institutional financial interests (no personal financial interests). The remaining authors declare no competing interests.

## Additional information

Philipp Heumann [1,2,33], Miguel E. Aguado-Barrera [3,4,33], Harkeran K. Jandu[5], David Azria[6], Erik Briers[34], Renée Bultijnck [7,8], Jenny Chang-Claude[1], Ananya Choudhury[9], Alison M. Dunning [10], Laura Fachal[11], Olivia Fuentes-Ríos[3,4], Antonio Gómez-Caamaño [4,12], Sara Gutiérrez-Enríquez [13], Rudolf Kaaks[1], Sarah L. Kerns[14], Maarten Lambrecht [15], Juan Camilo Rosas Romero[1], Barry S. Rosenstein[16], Dirk De Ruysscher[17], Martina E. Schmidt [18], Elena Sperk [19], Hilary Stobart[34], R. Paul Symonds[5], Liv Veldeman[8], Marlon R. Veldwijk [20], Tim Ward[34], Adam Webb [5], Catharine M. L. West[9], Tiziana Rancati [21], Tim Rattay[5], Ana Vega [3,4,22], REQUITE Consortium*, Christopher J. Talbot [5] & Petra Seibold [1,23] ✉

[1]Division of Cancer Epidemiology, German Cancer Research Center (DKFZ), Heidelberg, Germany. [2]Medical Faculty Heidelberg, Heidelberg University, Heidelberg, Germany. [3]Instituto de Investigación Sanitaria De Santiago de Compostela, Santiago de Compostela, Spain. [4]Fundación Pública Galega Medicina Xenómica, Santiago de Compostela, Santiago de Compostela, Spain. [5]University of Leicester, Leicester, United Kingdom. [6]Fédération Universitaire d'Oncologie Radiothérapie d'Occitanie Méditerranée, ICM, Univ Montpellier INSERM U1194 IRCM, Montpellier, France. [7]Ghent University, Department of Human Structure and Repair, Ghent, Belgium. [8]Ghent University Hospital, Ghent, Belgium. [9]University of Manchester, Christie Hospital NHS Foundation Trust, Manchester, United Kingdom. [10]University of Cambridge, Cambridge, United Kingdom. [11]Wellcome Sanger Institute, Hinxton, United Kingdom. [12]Hospital Clínico Universitario de Santiago de Compostela, Servizo Galego de Saúde (SERGAS), Santiago de Compostela, Santiago de Compostela, Spain. [13]Hereditary Cancer Genetics Group, Vall d'Hebron Institute of Oncology (VHIO), Vall d'Hebron Barcelona Hospital Campus, Barcelona, Spain. [14]Medical College of Wisconsin, Milwaukee, USA. [15]University Hospital Leuven, Leuven, Belgium. [16]Icahn School of Medicine at Mount Sinai, New York, USA. [17]MAASTRO Clinic, Maastricht, Netherlands. [18]Division of Physical Activity, Cancer Prevention and Survivorship, German Cancer Research Center (DKFZ), Heidelberg, Germany. [19]Mannheim Cancer Center, Medical Faculty Mannheim, Heidelberg University, Mannheim, Germany. [20]Department of Radiation Oncology, Universitätsmedizin Mannheim, Medical Faculty Mannheim, Heidelberg University, Mannheim, Germany. [21]Fondazione IRCCS Istituto Nazionale dei Tumori, Milan, Italy. [22]Centro de Investigación en Red de Enfermedades Raras (CIBERER), 28029 Madrid, Spain. [23]Division of Personalized Early Detection of Prostate Cancer, German Cancer Research Center (DKFZ), Heidelberg, Germany. [33]These authors contributed equally: Philipp Heumann, Miguel E. Aguado-Barrera. [34]Unaffiliated: Patient Advocate. *A list of authors and their affiliations appears at the end of the paper. ✉e-mail: p.seibold@dkfz-heidelberg.de

## REQUITE Consortium

Barbara Avuzzi[21], Patricia Calvo-Crespo[3,12], Ana Carballo[3,12], Alessandro Cicchetti[21], Gilles Defraene[24], Isabel Dominguez-Rios[12], Roxana Draghici[6], Irene Fajardo-Paneque[12], Juan Fernández-Tajes[3,4], Valérie Fonteyne[7,8], Pietro Gabriele[25], Ulrich Giesche[26], Karin Haustermans[24], Irmgard Helmbold[1], Carsten Herskind[20], Kiran Kancherla[27], Christopher Kent[27], Ramón Lobato-Busto[12], Sara Morlino[21], Piet Ost[7,28], Debbie Payne[29], Paula Peleteiro[3,12], Belinda Rodriguez-Lage[3,4], Rebecca M. Shearer[30], Paloma Sosa-Fajardo[12], Petra Stegmaier[31], Richard Stock[16], Holly Summersgill[9], Begoña Taboada-Valladares[3,12], Laura Torrado-Moya[12], Riccardo Valdagni[21,32], Ben Vanneste[17] & Subramaniam Vasanthan[27]

[24]KU Leuven, Leuven, Belgium. [25]Department of Radiation Oncology, Candiolo Cancer Institute - FPO, IRCCS, (TO), Candiolo, Italy. [26]Klinik für Strahlentherapie, St. Vincentius-Kliniken gAG, Karlsruhe, Germany. [27]University Hospitals of Leicester NHS Trust, Leicester, UK. [28]Department of Radiation Oncology, GZA Sint-Augustinus, Antwerpen, Belgium. [29]Centre for Integrated Genomic Medical Research (CIGMR), Manchester, UK. [30]The Christie NHS Foundation Trust, Manchester, UK. [31]Zentrum für Strahlentherapie Freiburg, Breisgau, Germany. [32]Department of Oncology and Haematology-Oncology, University of Milan, Milan, Italy.

