## [Transparent Peer Review file · Nature Communications]

Genetic determinants of fatigue up to 2 years after radiotherapy in prostate cancer patients

Corresponding Author: Dr Petra Seibold

Version 0:

Reviewer comments:

Reviewer #1

(Remarks to the Author)

Cancer related fatigue is highly prevalent and distressing and its biological determinants remain understudied. This study presents a GWAS analysis of long term cancer related fatigue in 1358 patients with non metastatic prostate cancer treated with radiotherapy from Europe and US included in a prospective cohort study. Using validated fatigue tools (EORTC C30 and in some centers also MFI), the authors suggest the identification of a genome wide significant association between a SNP in the ACTR3/CBWD2 region and physical fatigue in patients without pre-treatment fatigue. The strengths of the study include its multicentric and prospective nature and the use of validated fatigue tools, nevertheless there was a absence of validation cohort to support the findings.

Some methodological issues limit the interpretability of the findings :

1. Absence of validation/replication of the SNPs association in an independent cohort.
2. Although gene expression differences were observed for ACTR3 and CBWD2, no significant findings prevailed after correction for multiple testing.
3. Clarification why only a subset of patients were evaluated with MFI and what defined this subset of patients.
4. The term « Clinically important » fatigue in the abstract should be defined. Although in the methods the threshold for the C30 subscale use is appropriate, the MFI cut-off was data-driven and may not be clinically meaningful. This should be acknowledged and authors should consider perform sensitivity analyses.
4. Handling of missing data via mean imputation can introduce bias, particularly if the non-response is associated with the level of fatigue. Did the authors evaluate if missingness was random ? Consider sensitivity analyses.
5. The definition of fatigue is based on either Year 1 or 2 presence which is a long time frame. Should a sensitivity analyses requiring fatigue at both time points or just focusing on one of the time points should be considered ?
6. Comment on the imputation score threshold used. 0.3 may be considered permissive and allow for poorly imputed variants.
7. Was HWE filtering performed ?

Finally, I would recommend avoiding strong mechanistic interpretations unless expression differences are replicated and more robust, particularly claims regarding biological mechanisms and genetic associations should be more presented cautiously.

Reviewer #2

(Remarks to the Author)

What are the noteworthy results?

Fatigue in prostate cancer patients is common but has not been extensively studied. The present study is relatively large in cohort size, with links to possible genetical predisposition.

Will the work be of significance to the field and related fields?

Yes.

How does it compare to the established literature?

Large cohort with links between fatigue and possible genetic predisposition.

If the work is not original, please provide relevant references.
It can be regarded as original.

Does the work support the conclusions and claims, or is additional evidence needed?

Not completely. The evidence presented by the study may support increased fatigue after the diagnosis of prostate cancer in general. Additional evidence is needed to specifically support that radiation therapy can cause increased fatigue.

Is the methodology sound? Does the work meet the expected standards in your field?

In general, the methodology is sound and meet the expected standards in my field as a clinical oncologist both in medical and radiation therapy of prostate cancer.

Some important clinical comments:

1. In the general summary data table, there is no data summary of Gleason score and ISUP grade which is one of the major factors for risk classification. GS/ISUP should be given for all clinical studies of prostate cancer.

2. The authors should provide data of subgroup analyses:

1) RT plus ADT vs RT, because it is well-known that even short-term ADT can cause both short- and long-term fatigue. ADT may have stronger influence than RT in terms of fatigue.

2) PORT (prostate only RT) vs WPRT (whole pelvic RT), because WPRT has much larger irradiated volume and has much more possible bone marrow suppression than PORT.

3. Although it is impossible to carry out a direct comparison between patients with and without definite RT within the cohort in the study, the authors may discuss and even focus on why RT is regarded as the cause of increased fatigue in this study. In general, fatigue is common and usually increases for all patients after diagnosis of prostate cancer, may be even in patients under active surveillance without any active treatment.

Is there enough detail provided in the methods for the work to be reproduced?

If the above major concerns in point 2 and 3 can be addressed.

Reviewer #3

(Remarks to the Author)

Dear authors,

Thank you for the opportunity to review the manuscript entitled "Genetic determinants of fatigue up to 2 years after radiotherapy in prostate cancer patients". This study is novel, and the insights generated are hypothesis-generating and may open opportunities in the field of radiogenomics of fatigue. While the limitations of the modest sample size and phenotypic definitions are acknowledged, this study is still plagued by the study-design limitations of GWAS. Without an independent or external validation set, the findings of this study lack robustness, rendering the results preliminary. I acknowledge that the availability of an independent validation set is not always possible, particularly while addressing a novel research question.

My recommendation is, therefore, Major Revisions. I have outlined my suggestions below to enhance the robustness, interpretability, and transparency of your findings.

1. Internal validation and robustness checks

a. Bootstrap/Jackknife Analyses: The rationale is to assess the stability of the lead SNP effect, as well as any other suggestive signals under resampling. For the incident physical fatigue phenotype (subgroup without baseline fatigue; $n \approx 640$), perform bootstrap resampling (e.g., $\geq 1,000$ replicates). In each replicate, re-estimate the odds ratio (OR) and p-value for rs142212041. Report the distribution of effect estimates and confidence intervals from the bootstrap. If the OR remains consistently in the same direction with similar magnitude, it supports robustness despite a limited sample size.

b. Jackknife/Leave-One-Out: Particularly if REQUITE spans multiple centers/countries, perform leave-one-site-out analyses: exclude each center in turn, test direction of effect for the top SNP in the remaining data (even if underpowered, consistent direction across exclusions suggests a single site does not drive signal). Document results in a supplementary table.

c. Permutation testing: The rationale is to evaluate the empirical significance of the observed association given the analysis pipeline and phenotype distribution.

d. Sensitivity (analysis) to covariate specification using alternative covariate sets. E-fit the logistic model for the lead SNP with varied covariate sets: e.g., adding comorbidities (if available), physical activity or sleep variables (if collected), removing less certain covariates, and using different numbers of genetic ancestry PCs (e.g., first five vs. first 10). Report how the effect estimate and p-value change. Stability suggests robustness; if highly sensitive, discuss implications.

e. Fine-mapping and conditional analyses: Apply Bayesian fine-mapping tools (e.g., FINEMAP, CAVIAR) in the region around rs142212041 to derive a credible set of candidate causal variants. A small credible set focusing on rs142212041 (or a few variants) increases interpretability. Within ~ 1 Mb around the lead SNP, perform conditional association adjusting for

rs142212041 to test for additional independent signals. If none remain, the locus likely harbors a single signal; discuss this in the Results section.

2. In Silico supportive evidence via external datasets.

- a. Look up in the public GWAS of related traits. Although the specific phenotype in question may not have been previously reported, summary statistics for other related fatigue phenotypes may be available (e.g., UK Biobank).
- b. If GWAS summary statistics for a related trait (e.g., general fatigue, depression) are available, compute genetic correlation (r_g) with your fatigue GWAS summary stats. A positive r_g suggests shared genetic architecture, lending support to the validity of your phenotype and the plausibility of the signals.
- c. Use publicly available eQTL weights (e.g., from GTEx) to perform TWAS on your GWAS summary statistics. Focus on candidate genes in the locus (ACTR3, CBWD2) in relevant tissues (blood, brain regions, muscle). A TWAS signal that aligns with your GWAS peak strengthens functional plausibility.
- d. Apply tools like COLOC to test whether the GWAS signal and eQTL signal for ACTR3/CBWD2 (or other nearby genes) in relevant tissues share the same causal variant. Positive colocalization supports the hypothesis that gene expression modulation contributes to the risk of fatigue.
- e. Annotate rs142212041 and proxies (e.g., via ANNOVAR, HaploReg, RegulomeDB) to determine overlap with enhancers, promoters, or transcription factor binding sites active in tissues/cell types implicated in fatigue (e.g., immune cells, muscle, brain). Report any enrichment in relevant regulatory marks (e.g., H3K27ac peaks) using ENCODE/Roadmap data.
- f. Conduct gene-based or pathway enrichment analyses (e.g., MAGMA, DEPICT) on full GWAS summary statistics. If pathways related to cytoskeleton, extracellular vesicle biology, immune signaling, or energy metabolism emerge, this convergence adds weight to your hypothesis-generating finding.

3. Methodology clarification:

- a. Provide rationale for genotype imputation thresholds (e.g., INFO \geq 0.3), MAF cut-offs, and missingness filters.
- b. Describe how missing covariate or phenotype data were handled (e.g., exclusion, imputation).
- c. Justify the selection of covariates by including a table that lists all variables considered and the final model covariates.

4. Expand limitations:

- a. Explicitly acknowledge limited power to detect small-effect variants, wide CIs, and the preliminary nature of findings without replication.
- b. Discuss the subjectivity of fatigue assessments, implications of dichotomization, and potential misclassification.
- c. Note analyses restricted to primarily European ancestry limit applicability to other populations.
- d. Acknowledge potential influences of comorbidities, lifestyle factors, and psychosocial variables not fully captured.
- e. Discuss that whole-blood eQTL may not reflect relevant tissues (CNS, muscle) and that in silico eQTL from GTEx have limitations (post-mortem tissue, older donors).

5. Data and code availability:

- a. Commit to publicly sharing de-identified summary statistics (e.g., via GWAS Catalog or institutional repository) to enable future meta-analyses.
- b. Make analysis pipelines (QC, association tests, resampling scripts, phenotype derivation) available (e.g., GitHub) to enhance reproducibility.

6. Interpretation and discussion:

- a. Reframe statements about the lead association as “hypothesis-generating” rather than definitive. For example: “We identify a locus near ACTR3/CBWD2 that merits further investigation; replication is required to confirm this signal.”
- b. Avoid overemphasis on biological interpretation before in silico and functional corroboration; instead, present the mechanistic links as plausible but unconfirmed.
- c. Highlight how in silico annotations (eQTL, regulatory overlap), pathway enrichment, and phenotype correlations (PRS/genetic correlation) together suggest biological plausibility, even if each line of evidence alone is insufficient.
- d. Discuss how these convergent data justify prioritizing this locus for future functional studies.
- e. Explain what SNP-based heritability estimates (e.g., $H^2 \approx 0.31 \pm 0.18$ for overall fatigue) imply about genetic contribution vs. environmental/noise, comparing with prior heritability estimates for fatigue or related traits.
- f. Discuss limitations of heritability estimation (e.g., measurement error, sample size).
- g. Outline concrete next steps: forming collaborations for meta-analysis, collecting similar phenotype data in other cohorts, functional studies (in vitro/in vivo), and exploring intervention targets if findings hold.
- h. Emphasize that while replication is ideal, the current analyses (robustness checks, in silico triangulation) build a foundation for prioritizing loci and guiding experimental validation.

Version 1:

Reviewer comments:

Reviewer #1

(Remarks to the Author)

Overall, the manuscript is greatly improved after revision. Regarding the points I previously raised, they have all been addressed with the exception of the issue concerning missingness. The response suggests stability of one estimate but does not answer the reviewer's question about the randomness of missingness, justify the use of mean imputation nor does it

include optimal sensitivity analyses. A complete case re-analysis is not a valid test of whether data are missing completely at random, missing at random or missing not at random and therefore cannot address potential bias arising from outcome dependent dropout. Ideally, the authors should have considered: 1 - A description and modeling of missingness (including whether baseline fatigue predicts loss), 2 - A primary analysis that avoids mean imputation (e.g. mixed-effects maximum likelihood or multiple imputation), and 3- Sensitivity analyses using inverse probability weighting (IPW) and MNAR (delta-adjusted) approaches for the lead SNPs.

Reviewer #3

(Remarks to the Author)

Dear authors, thank you for your comprehensive responses to the comments raised. I am satisfied that all my comments have been addressed.

Version 2:

Reviewer comments:

Reviewer #1

(Remarks to the Author)

The authors have addressed the questions and clarified the key points.

RESPONSE TO REVIEWERS' COMMENTS

Thank you for giving us the opportunity to revise our manuscript based on the positive feedback and thoughtful comments from all three reviewers. We have adapted the manuscript accordingly to reflect the reviewers' comments and added the suggested comprehensive additional analyses. All changes in the manuscript were added in editing mode for easy identification. We sincerely thank the reviewers for their constructive feedback, which substantially improved the manuscript.

Below you will find a point-by-point list of the changes we have made:

Reviewer 1

1. *“Clarification why only a subset of patients were evaluated with MFI and what defined this subset of patients.”*

We have added the following information to the methods section to clarify that Multidimensional Fatigue Inventory (MFI) data were collected only in a subset of the study centres due to logistic reasons (page 4, line 126):

“In some centres in six out of eight countries, patient-reported fatigue was additionally assessed using the Multidimensional Fatigue Inventory (MFI) at the same time points [16].”

2. *“The term « Clinically important » fatigue in the abstract should be defined. Although in the methods the threshold for the C30 subscale use is appropriate, the MFI cut-off was data-driven and may not be clinically meaningful. This should be acknowledged and authors should consider perform sensitivity analyses.”*

We have added the following information to the abstract (page 2, lines 76-78):

“Clinically important fatigue was defined for the EORTC QLQ-C30 based on the proposed threshold as scores ≥ 39 and for the MFI as scores ≥ 75 th percentile in the general German male population aged ≥ 60 years.”

To assess the impact of the threshold values chosen for the MFI, we repeated the genome-wide association analysis for incident physical fatigue, raising the threshold for clinically important fatigue from scores ≥ 75 th percentile to scores ≥ 90 th percentile in the general German male population aged ≥ 60 years. Using a stricter threshold, thereby reducing the statistical power by

lowering the proportion of men classified as fatigued, the effect estimate for the index SNP rs142212041 still remains comparable in direction and magnitude with an odds ratio of 2.26 (95% confidence interval: 1.42 to 3.61).

We have added the following information to the methods section (page 5, lines 180-183):

“Additionally, the threshold for clinically important fatigue was raised from scores ≥ 75 th percentile to scores ≥ 90 th percentile in the general German male population aged ≥ 60 years to examine a more stringent threshold in fatigue classification.”

We have added the following information to the results section (page 9, lines 308-309):

“The effect estimates for rs142212041 remained consistent in both direction and magnitude across subgroup and sensitivity analyses (Supplementary table IV).”

We have added the following table to the Appendix, which summarises the results of several additional subgroup and sensitivity analyses (page 49; highlighting of the presented analysis for this reply):

Supplementary table IV. Odds ratios and 95% confidence intervals for the index SNP rs142212041 in the genome-wide association subgroup and sensitivity analyses on clinically important incident physical fatigue in the MFI up to two years after the end of radiotherapy in prostate cancer patients of the REQUITE cohort without clinically important fatigue before radiotherapy.

	Odds Ratio (95% Confidence Interval) rs142212041
Subgroup Analyses[†]	
Receipt of ADT	
No ADT (N=207)	2.53 (1.32-4.84)
Any ADT (N=436)	3.52 (2.15-5.76)
Receipt of Pelvic Radiotherapy	
No Pelvic Radiotherapy (N=397)	3.20 (2.01-5.10)
Pelvic Radiotherapy (N=246)	3.09 (1.44-6.65)
Time Point of Fatigue Assessment	

1 Year Post-Radiotherapy (N=603)	2.05 (1.30-3.23)
2 Years Post-Radiotherapy (N=548)	3.29 (2.08-5.20)
More Restrictive Threshold of Clinical Importance² (N=755)	2.26 (1.42-3.61)
Complete MFI Physical Fatigue Assessments (N=620)	3.05 (2.04-4.55)
Sensitivity Analyses¹	
Reduced Covariate Set³ (N=643)	2.92 (1.99-4.29)
Extended by five additional genetic ancestry principal components (N=643)	2.98 (2.02-4.42)
¹ N corresponds to the number of patients included in the regression model. ² The threshold for clinically important fatigue was raised from scores ≥ 75th to scores ≥ 90th percentile in the general German male population aged ≥ 60 years [24]. ³ The reduced covariate set was derived by fitting a logistic regression model with clinically significant fatigue up to two years after radiotherapy adjusted for demographic, disease and treatment-specific covariates (excluding genetic data) as previously described. Covariates with $P \leq 0.2$ were included in the reduced covariate set.	

To emphasise the lack of a validated threshold for the MFI as a limitation of the analysis, we have made the following change in the discussion section (page 20, lines 484-487, 494-496):

“The presented work has some limitations. (...) As no validated threshold for clinically important fatigue was previously published for the MFI, patients were classified as fatigued and non-fatigued using population-based reference values, as opposed to sample-based cut-offs. (...) ~~As no validated threshold for clinically important fatigue was available for the MFI, patients were classified as fatigued and non-fatigued using population-based reference values, as opposed to sample-based cut-offs.~~”

3. *“Handling of missing data via mean imputation can introduce bias, particularly if the non-response is associated with the level of fatigue. Did the authors evaluate if missingness was random ? Consider sensitivity analyses.”*

As a subgroup analysis, we repeated the genome-wide association analysis for incident physical fatigue in the MFI with 620 patients for whom complete MFI data were available for the physical fatigue dimension before the start of radiotherapy and up to two years later. With an odds ratio of 3.05 (95% confidence interval: 2.04 to 4.55), the observed effect estimate for index SNP rs142212041 differed only slightly from that for the cohort of men including mean imputed assessments (OR: 2.99; 95% confidence interval: 2.03 to 4.41). In this complete case analysis, the SNP rs7066898 was also genome-wide statistically significantly associated with incident physical fatigue. The results were visualised using a Manhattan plot and included in the Appendix.

We added the following information to the methods section (page 5, lines 183-184):

“Non-responder bias was investigated by restricting the analysis to patients with complete fatigue assessments.”

We have added the following information to the results section (page 9, lines 308-312):

“The effect estimates for rs142212041 remained consistent in both direction and magnitude across subgroup and sensitivity analyses (Supplementary table IV). Limiting the analysis to men with complete MFI assessments, a genome-wide statistically significant association with incident physical fatigue was also observed for rs7066898 (OR: 1.79; 95% confidence interval: 1.45 to 2.21; $P=5\times 10^{-8}$; Supplementary figure C).”

The results have been added to Supplementary table IV (see previous comment 2), which summarises the results of several subgroup and sensitivity analyses (page 49).

We have added the following figure to the Appendix (page 50):

Supplementary figure C. Manhattan plot for physical fatigue in the Multidimensional Fatigue Inventory (N=620) up to two years following external beam radiotherapy in prostate cancer patients of the REQUITE cohort without fatigue before the start of radiotherapy who provided complete MFI assessments for the physical fatigue dimension. Multivariable regression models were adjusted for age and BMI at enrolment, baseline fatigue level prior to radiotherapy, depression, hormone therapy, prostatectomy, tumour size, nodal involvement, pelvic radiotherapy, and the top 5 principal components.

4. *“The definition of fatigue is based on either Year 1 or 2 presence which is a long time frame. Should a sensitivity analyses requiring fatigue at both time points or just focusing on one of the time points should be considered ?”*

To evaluate the impact of the time point of the long-term fatigue assessment, the genome-wide association analysis for incident physical fatigue in the MFI was repeated separately for fatigue assessments at one year and at two years after radiotherapy. For fatigue one year after radiotherapy, an odds ratio of 2.05 (95 % confidence interval: 1.30 to 3.23, N=603) was observed for the index SNP rs1422129412. Fatigue two years after radiotherapy gave an odds ratio of 3.29

(95% confidence interval: 2.08 to 5.20, N=548). The effect estimates do not indicate that rs142212041 is exclusively associated with fatigue at either time point. To maximise the sample size, we have used the combined endpoint at either one year or two years for the main analysis.

We have added the following information to the methods section (page 5, lines 178-180):

“As subgroup analyses, the genome-wide association analysis was stratified by receipt of ADT and pelvic radiotherapy and repeated separately for fatigue assessments at one year or at two years after radiotherapy.”

We have added the following information to the results section (page 9, lines 308-309):

“The effect estimates for rs142212041 remained consistent in both direction and magnitude across subgroup and sensitivity analyses (Supplementary table IV).”

The results have been added to Supplementary table IV (see previous comment 2), which summarises the results of several subgroup and sensitivity analyses (page 49).

5. *“Comment on the imputation score threshold used. 0.3 may be considered permissive and allow for poorly imputed variants.”*

While the selected imputation score threshold of 0.3 may be considered permissive, it was chosen in consideration of previously published genome-wide association studies on radiotherapy toxicities in prostate cancer [1,2] and breast cancer patients [3]. In view of the exploratory nature of the analysis emphasised in the manuscript and to facilitate comparability with published analyses, we suggest retaining the chosen threshold value.

We have added the following information to the methods section (page 4, lines 151-152; see also Reviewer 3, comment 3a):

“To facilitate comparability of findings with previous studies on toxicities after radiotherapy, the analysis was based on common SNPs with MAF ≥ 0.05 and imputation score ≥ 0.3 [22-24].”

6. *“Was HWE filtering performed ?”*

HWE filtering was performed in advance.

We have added the following information to the methods section (page 4, lines 147-148):

“2,746 variants were excluded due to deviations from Hardy-Weinberg Equilibrium ($P < 10^{-7}$).”

7. *“Finally, I would recommend avoiding strong mechanistic interpretations unless expression differences are replicated and more robust, particularly claims regarding biological mechanisms and genetic associations should be more presented cautiously.”*

We have reviewed the manuscript to avoid strong mechanistic interpretations. To highlight that the results presented are initial evidence requiring further replication, we have made the following addition to the discussion section (page 20, lines 509-510):

“In conclusion, this study emphasizes the multi-dimensionality of fatigue and has provided initial evidence for its heritability across different fatigue phenotypes.”

Furthermore, we have included the following statement in the discussion section (page 20, lines 499-501):

“It is still unclear whether the findings can be generalised to patients with metastatic disease at diagnosis, other tumour entities, or populations of non-European ancestry.”

Reviewer 2

1. *“In the general summary data table, there is no data summary of Gleason score and ISUP grade which is one of the major factors for risk classification. GS/ISUP should be given for all clinical studies of prostate cancer.”*

We have added the ISUP grade in Table 1 (page 11-12) and in Supplementary table I (page 32-34).

Table 1. Selected characteristics of prostate cancer patients with available genetic data from the REQUITE cohort that assessed the EORTC QLQ-C30 Fatigue Scale and at least one Multidimensional Fatigue Inventory dimension in a sub-cohort up to two years following external beam radiotherapy.

EORTC QLQ-C30 Fatigue Scale		Multidimensional Fatigue Inventory
Characteristic	N = 1,358 ¹	N = 869 ¹
...
ISUP Grade		
1 (Gleason score: 2–6)	229 (17%)	164 (19%)
2 (Gleason score: 3+4)	444 (33%)	283 (33%)
3 (Gleason score: 4+3)	336 (25%)	235 (27%)
4 (Gleason score: 8)	168 (12%)	92 (11%)
5 (Gleason score: 9–10)	176 (13%)	93 (11%)
...

Supplementary table I. Selected characteristics of prostate cancer patients with available genetic data from the REQUITE cohort that assessed the EORTC QLQ-C30 Fatigue Scale and/or at least one Multidimensional Fatigue Inventory dimension up to two years following external beam radiotherapy stratified by clinically important fatigue levels before the start of radiotherapy

EORTC QLQ-C30			Multidimensional Fatigue Inventory									
Fatigue Scale			General Fatigue		Physical Fatigue		Mental Fatigue		Reduced Activity		Reduced Motivation	
Characteristic	N = 1,284 ¹		N = 822 ¹		N = 821 ¹		N = 817 ¹		N = 820 ¹		N = 819 ¹	
	No clinically important fatigue prior to RT (N = 1,147) ¹	Clinically important fatigue prior to RT (N = 137) ¹	No clinically important fatigue prior to RT (N = 677)	Clinically important fatigue prior to RT (N = 149) ¹	No clinically important fatigue prior to RT (N = 643)	Clinically important fatigue prior to RT (N = 182) ¹	No clinically important fatigue prior to RT (N = 666) ¹	Clinically important fatigue prior to RT (N = 155) ¹	No clinically important fatigue prior to RT (N = 663) ¹	Clinically important fatigue prior to RT (N = 161)	No clinically important fatigue prior to RT (N = 676)	Clinically important fatigue prior to RT (N = 147)
...
ISUP Grade												
1 (Gleason score: 2–6)	199 (17%)	19 (14%)	139 (21%)	20 (14%)	122 (19%)	37 (20%)	130 (20%)	28 (18%)	128 (19%)	31 (19%)	128 (19%)	31 (21%)
2 (Gleason score: 3+4)	372 (33%)	49 (36%)	222 (33%)	47 (32%)	211 (33%)	58 (32%)	215 (32%)	52 (34%)	222 (34%)	47 (29%)	229 (34%)	40 (27%)
3 (Gleason score: 4+3)	295 (26%)	25 (18%)	190 (28%)	33 (22%)	180 (28%)	42 (23%)	180 (27%)	42 (27%)	183 (28%)	39 (24%)	186 (28%)	35 (24%)
4 (Gleason score: 8)	138 (12%)	18 (13%)	64 (9%)	24 (16%)	65 (10%)	23 (13%)	75 (11%)	12 (8%)	68 (10%)	19 (12%)	68 (10%)	19 (13%)
5 (Gleason score: 9–10)	139 (12%)	26 (19%)	61 (9%)	24 (16%)	63 (10%)	22 (12%)	64 (10%)	21 (14%)	60 (9%)	25 (16%)	64 (9%)	21 (14%)
...

2. *“The authors should provide data of subgroup analyses:*

1) RT plus ADT vs RT, because it is well-known that even short-term ADT can cause both short- and long-term fatigue. ADT may have stronger influence than RT in terms of fatigue.

2) PORT (prostate only RT) vs WPRT (whole pelvic RT), because WPRT has much larger irradiated volume and has much more possible bone marrow suppression than PORT.”

We performed both proposed subgroup analyses by repeating the genome-wide association analysis on incident physical fatigue in the MFI stratified by receipt of ADT (none vs. any) and receipt of radiotherapy to the pelvis (none vs. any). In men treated with ADT (OR: 3.52; 95% confidence interval: 2.15 to 5.76) and without ADT (OR: 2.53; 95% confidence interval: 1.32 to 4.84) as well as in men treated with pelvic radiotherapy (OR: 3.09; 95% confidence interval: 1.44 to 6.65) and without pelvic radiotherapy (OR: 3.20; 95% confidence interval: 2.01 to 5.10), positive associations of comparable magnitude were observed between the index SNP rs142212041 and incident physical fatigue.

We have added the following information to the methods section (page 5, lines 178-180):

“As subgroup analyses, the genome-wide association analysis was stratified by receipt of ADT and pelvic radiotherapy and repeated separately for fatigue assessments at one year or at two years after radiotherapy.”

We have added the following table to the Appendix, which summarises the results of several subgroup and sensitivity analyses (highlighting of the presented analysis for this reply; page 49):

Supplementary table IV. Odds ratios and 95% confidence intervals for the index SNP rs142212041 in the genome-wide association subgroup and sensitivity analyses on clinically important incident physical fatigue in the MFI up to two years after the end of radiotherapy in prostate cancer patients of the REQUITE cohort without clinically important fatigue before radiotherapy.

	Odds Ratio (95% Confidence Interval) rs142212041
Subgroup Analyses¹	
Receipt of ADT	
No ADT (N=207)	2.53 (1.32-4.84)

Any ADT (N=436)	3.52 (2.15-5.76)
Receipt of Pelvic Radiotherapy	
No Pelvic Radiotherapy (N=397)	3.20 (2.01-5.10)
Pelvic Radiotherapy (N=246)	3.09 (1.44-6.65)
Time Point of Fatigue Assessment	
1 Year Post-Radiotherapy (N=603)	2.05 (1.30-3.23)
2 Years Post-Radiotherapy (N=548)	3.29 (2.08-5.20)
More Restrictive Threshold of Clinical Importance² (N=755)	2.26 (1.42-3.61)
Complete MFI Physical Fatigue Assessments (N=620)	3.05 (2.04-4.55)
Sensitivity Analyses¹	
Reduced Covariate Set³ (N=643)	2.92 (1.99-4.29)
Extended by five additional genetic ancestry principal components (N=643)	2.98 (2.02-4.42)
¹ N corresponds to the number of patients included in the regression model. ² The threshold for clinically important fatigue was raised from scores ≥ 75th to scores ≥ 90th percentile in the general German male population aged ≥ 60 years [24]. ³ The reduced covariate set was derived by fitting a logistic regression model with clinically significant fatigue up to two years after radiotherapy adjusted for demographic, disease and treatment-specific covariates (excluding genetic data) as previously described. Covariates with $P \leq 0.2$ were included in the reduced covariate set.	

3. *“Although it is impossible to carry out a direct comparison between patients with and without definite RT within the cohort in the study, the authors may discuss and even focus on why RT is regarded as the cause of increased fatigue in this study. In general, fatigue is common and usually increases for all patients after diagnosis of prostate cancer, may be even in patients under active surveillance without any active treatment.”*

We acknowledge that the available data do not allow us to establish the cause of fatigue. Although the cohort in the secondary analysis was restricted to men without fatigue before radiotherapy, fatigue up to two years later may be non-radiation-related and thus bias the results.

We have added the following information to the discussion (page 19, lines 475-477; page 20, lines 478-484):

“Although the genome-wide statistically significant SNP for incident physical fatigue was observed in the subgroup of men unfatigued prior to radiotherapy, this finding could be biased by non-radiation related manifestation of fatigue. The available data do not allow to establish radiotherapy as causative for the observed fatigue, limiting the interpretation of the findings. However, the genome-wide statistically significant SNP for physical fatigue was observed in the subgroup of men without fatigue prior to radiotherapy after adjustment for demographic, disease- and treatment-specific characteristics. Therefore, it might be assumed that the fatigue that developed during radiotherapy is largely related to the treatment, although other factors such as anxiety, stress, pain, or insomnia due to the cancer diagnosis may have contributed.”

Reviewer 3

1. *“Internal validation and robustness checks*

- a. *Bootstrap/Jackknife Analyses: The rationale is to assess the stability of the lead SNP effect, as well as any other suggestive signals under resampling. For the incident physical fatigue phenotype (subgroup without baseline fatigue; n≈640), perform bootstrap resampling (e.g., ≥1,000 replicates). In each replicate, re-estimate the odds ratio (OR) and p-value for rs142212041. Report the distribution of effect estimates and confidence intervals from the bootstrap. If the OR remains consistently in the same direction with similar magnitude, it supports robustness despite a limited sample size.”*
- b. *Jackknife/Leave-One-Out: Particularly if REQUITE spans multiple centers/countries, perform leave-one-site-out analyses: exclude each center in turn, test direction of effect for the top SNP in the remaining data (even if underpowered, consistent direction across exclusions suggests a single site does not drive signal). Document results in a supplementary table.*
- c. *Permutation testing: The rationale is to evaluate the empirical significance of the observed association given the analysis pipeline and phenotype distribution.*
- d. *Sensitivity (analysis) to covariate specification using alternative covariate sets. E-fit the logistic model for the lead SNP with varied covariate sets: e.g., adding comorbidities (if available), physical activity or sleep variables (if collected), removing less certain covariates, and using different numbers of genetic ancestry PCs (e.g., first five vs. first 10). Report how the effect estimate and p-value change. Stability suggests robustness; if highly sensitive, discuss implications.*
- e. *Fine-mapping and conditional analyses: Apply Bayesian fine-mapping tools (e.g., FINEMAP, CAVIAR) in the region around rs142212041 to derive a credible set of candidate causal variants. A small credible set focusing on rs142212041 (or a few variants) increases interpretability. Within ~1 Mb around the lead SNP, perform conditional association adjusting for rs142212041 to test for additional independent signals. If none remain, the locus likely harbors a single signal; discuss this in the Results section.”*

To assess the robustness of the SNP effects, bootstrap odds ratio and standard errors were calculated for the top 20 SNPs using 3,000 replicates and the parametric bootstrap approach proposed by Forde et al. [4]. 95% confidence intervals were computed using conditional likelihood methods as proposed by Ghosh et al. and implemented in the R package winnerscurse [5]. For

the index SNP rs142212041, the bootstrap odds ratio of 1.86 (standard error: 0.20) is consistent in direction with the non-bootstrap estimate (OR: 2.99), albeit of lower magnitude. The 95% confidence interval ranges from 1.01 to 3.37.

In a leave-one-out analyses at country level, the genome-wide association analysis was repeated by omitting one country respectively. The results suggest that no country exclusively drives the signal for the index SNP rs142212041.

As sensitivity analyses, a reduced covariate set and the inclusion of five additional genetic ancestry principal components were investigated as proposed. The reduced covariate set was derived by fitting a logistic regression model for incident physical fatigue up to two years after radiotherapy, adjusting for the covariates (excluding genetic data) listed in the methods section. Covariates with $P \leq 0.2$ (age, BMI, tumour size) were included in the reduced covariate set. The effect estimates for the index SNP remained consistent in both sensitivity analyses.

The genome-wide association analysis for incident physical fatigue in the MFI was repeated adjusted for the index SNP rs142212041. The conditional analysis did not indicate secondary signals in the genomic region around the index SNP.

Genetic fine-mapping was performed using the Sum of Single Effects model as outlined by Zou and implemented in the R package susieR [6]. 95% credible sets were determined in the susceptibility locus on chromosome 2 from 114.40 Mb to 114.80 Mb. Two 95% credible sets with a total of three candidate causal variants were derived; the first one containing rs35060225, the second one containing rs34241561 and rs12472252. Candidate causal variants were included in the SNP annotation (see comment 2).

We have added the following information to the methods section (page 5, lines 184-186; page 6, lines 199-206):

Association analysis
“As sensitivity analyses, the association analysis was repeated with a reduced covariate set and extended by five additional genetic ancestry principal components.”

Internal validation and fine-mapping
“Bootstrap odds ratios and standard errors were calculated using a parametric bootstrap approach and 3,000 replicates [32]. 95% confidence intervals were computed using conditional

likelihood methods as implemented in the R package winnerscurse. In a leave-one-out analyses at country level, the genome-wide association analysis was repeated by omitting one country respectively. Additional independent signals were examined in conditional analyses adjusted for genome-wide statistically significant SNPs. Fine-mapping was conducted using the Sum of Single Effects model and the package susieR [33].”

We have added the following information to the results section (page 9, lines 308-309, 312-316, 325-328):

“The effect estimates for rs142212041 remained consistent in both direction and magnitude across subgroup and sensitivity analyses (Supplementary table IV). (...) The bootstrap odds ratio of 1.86 for the index SNP rs142212041 (standard error: 0.20; 95% confidence interval: 1.01 to 3.37) is consistent in direction with the non-bootstrap estimate, albeit of lower magnitude (Supplementary figure D). The effect estimates of the leave-one-out analyses suggest that no country exclusively drives the signal for the index SNP (Supplementary table V). (...) The conditional analysis did not indicate secondary signals in the genomic region around the index SNP rs142212041. Fine-mapping of the susceptibility locus on chromosome 2 from 114.40 Mb to 114.80 Mb revealed two credible sets containing three putative causal variants: rs35060225; rs34241561 and rs12472252 (Supplementary figure E).”

We have added the following table to the Appendix, which summarises the results of several subgroup and sensitivity analyses (page 49; highlighting of the presented analysis for this reply):

Supplementary table IV. Odds ratios and 95% confidence intervals for the index SNP rs142212041 in the genome-wide association subgroup and sensitivity analyses on clinically important incident physical fatigue in the MFI up to two years after the end of radiotherapy in prostate cancer patients of the REQUITE cohort without clinically important fatigue before radiotherapy.

	Odds Ratio (95% Confidence Interval) rs142212041
Subgroup Analyses¹	
Receipt of ADT	
No ADT (N=207)	2.53 (1.32-4.84)

Any ADT (N=436)	3.52 (2.15-5.76)
Receipt of Pelvic Radiotherapy	
No Pelvic Radiotherapy (N=397)	3.20 (2.01-5.10)
Pelvic Radiotherapy (N=246)	3.09 (1.44-6.65)
Time Point of Fatigue Assessment	
1 Year Post-Radiotherapy (N=603)	2.05 (1.30-3.23)
2 Years Post-Radiotherapy (N=548)	3.29 (2.08-5.20)
More Restrictive Threshold of Clinical Importance² (N=755)	2.26 (1.42-3.61)
Complete MFI Physical Fatigue Assessments (N=620)	3.05 (2.04-4.55)
Sensitivity Analyses¹	
Reduced Covariate Set³ (N=643)	2.92 (1.99-4.29)
Extended by five additional genetic ancestry principal components (N=643)	2.98 (2.02-4.42)
¹ N corresponds to the number of patients included in the regression model. ² The threshold for clinically important fatigue was raised from scores ≥ 75th to scores ≥ 90th percentile in the general German male population aged ≥ 60 years [24]. ³ The reduced covariate set was derived by fitting a logistic regression model with clinically significant fatigue up to two years after radiotherapy adjusted for demographic, disease and treatment-specific covariates (excluding genetic data) as previously described. Covariates with $P \leq 0.2$ were included in the reduced covariate set.	

We have added the following figure to the Appendix, which visualises the GWAS and bootstrap effect estimates of the top 20 SNPs (page 51):

Supplementary figure D. Bootstrap (3,000 replicates) and GWAS-derived odds ratios and 95% confidence intervals for the top 20 SNPs in the genome-wide association analyses on clinically important incident physical fatigue in the MFI up to two years after the end of radiotherapy in prostate cancer patients of the REQUITE cohort without clinically important fatigue before radiotherapy.

We have added the following table to the Appendix, which summarises the results of the leave-one-out analyses (page 52):

Supplementary table V. Odds ratios and 95% confidence intervals for the SNP rs142212041 in the genome-wide association analyses on clinically important incident physical fatigue in the MFI up to two years after the end of radiotherapy in prostate cancer patients of the REQUITE cohort without clinically important fatigue before radiotherapy excluding one country respectively.

	Odds Ratio (95% Confidence Interval) rs142212041
Leave-One-Out Analysis¹	
Excluding France (N=487)	2.81 (1.80-4.38)
Excluding Germany (N=593)	3.44 (2.29-5.16)
Excluding Italy (N=524)	3.04 (1.99-4.63)
Excluding Spain (N=464)	2.30 (1.39-3.81)
Excluding United Kingdom (N=509)	3.22 (2.10-4.92)
Excluding United States (N=638)	2.96 (2.01-4.37)
¹ N corresponds to the number of patients included in the regression model.	

We have added the following figure to the Appendix, which visualises the genomic position of the candidate causal variants (page 53):

Supplementary figure E. Locus plot for the interval spanning 114.40 Mb to 114.80 Mb on chromosome 2 highlighting the genomic position of the candidate causal variants.

2. *"In Silico supportive evidence via external datasets.*

a. *Look up in the public GWAS of related traits. Although the specific phenotype in question may not have been previously reported, summary statistics for other related fatigue phenotypes may be available (e.g., UK Biobank).*

b. *If GWAS summary statistics for a related trait (e.g., general fatigue, depression) are available, compute genetic correlation (r_g) with your fatigue GWAS summary stats. A positive R_g suggests shared genetic architecture, lending support to the validity of your phenotype and the plausibility of the signals.*

c. Use publicly available eQTL weights (e.g., from GTEx) to perform TWAS on your GWAS summary statistics. Focus on candidate genes in the locus (*ACTR3*, *CBWD2*) in relevant tissues (blood, brain regions, muscle). A TWAS signal that aligns with your GWAS peak strengthens functional plausibility.

d. Apply tools like COLOC to test whether the GWAS signal and eQTL signal for *ACTR3*/*CBWD2* (or other nearby genes) in relevant tissues share the same causal variant. Positive colocalization supports the hypothesis that gene expression modulation contributes to the risk of fatigue.

e. Annotate *rs142212041* and proxies (e.g., via ANNOVAR, HaploReg, RegulomeDB) to determine overlap with enhancers, promoters, or transcription factor binding sites active in tissues/cell types implicated in fatigue (e.g., immune cells, muscle, brain). Report any enrichment in relevant regulatory marks (e.g., H3K27ac peaks) using ENCODE/Roadmap data.

f. Conduct gene-based or pathway enrichment analyses (e.g., MAGMA, DEPICT) on full GWAS summary statistics. If pathways related to cytoskeleton, extracellular vesicle biology, immune signaling, or energy metabolism emerge, this convergence adds weight to your hypothesis-generating finding.”

The shared genetic architecture between incident physical fatigue and EHR-derived malaise and fatigue and chronic fatigue syndrome (ME/CFS) in white British participants from the UK Biobank was investigated using publicly available GWAS data [7,8]. Local genetic correlation r_g was calculated for chromosome 2 as proposed by Werme et al. and implemented in the R package LAVA [9]. The 1000 Genomes Project (EUR) Phase 3 was used as a LD reference. Non-associated loci were filtered using univariate analysis and a Bonferroni-corrected significance level threshold of $\alpha=0.05/220=0.00023$. Bivariate tests were performed for the remaining loci.

For malaise and fatigue (3,413 cases, 403,789 controls), bivariate tests were calculated for 158 loci on chromosome 2 with a mean r_g of 0.03. 19 loci showed a Bonferroni-corrected statistically significant bivariate r_g , including the locus around the lead SNP *rs142212041* ($r_g=-0.09$, $P=0.0001$).

For ME/CFS (590 cases, 403,791 controls), bivariate tests were calculated for 158 loci, 18 of which showed a Bonferroni-corrected statistically significant bivariate r_g . Across loci on chromosome 2, mean r_g was 0.03. For the locus harbouring the leading SNP, a highly statistically significant local genetic correlation between incident physical fatigue and chronic fatigue syndrome $r_g=0.18$ was observed ($P=5.79\times 10^{-13}$). Statistically significant local genetic correlations

between fatigue in prostate cancer patients and other fatigue phenotypes in the susceptibility locus suggest potentially shared genetic basis supporting the plausibility of the GWAS signal.

Colocalisation was explored using multi-tissue expression quantitative trait loci (eQTL) data derived from brain (hippocampus), pituitary gland, and skeletal muscle tissue and whole blood (Genotype-Tissue Expression (GTEx) project v7) [10]. The analysis was conducted using the R package eQTLPlot [11]. In the locus 200 kb around *ACTR3*, the *P*-values of the genome-wide association analysis for incident physical fatigue and cis-eQTLs correlated most strongly for skeletal muscle tissue ($r=0.79$, $P=0.02$; Supplementary Figure F) and whole blood ($r=0.63$, $P=0.10$; Supplementary figure G). No significant associations were observed for pituitary gland ($r=0.47$, $P=0.21$; Supplementary figure H) and hippocampal tissue ($r=-0.17$, $P=0.89$; Supplementary figure I). Across tissues, no evidence was found that the trait-significant variant rs142212041 was associated with *ACTR3* expression levels. However, GTEx samples were collected from deceased subjects, which may affect expression data [12].

SNP annotation was performed for the lead SNP rs142212041, candidate causal variants rs35060225, rs34241561, rs12472252, and variants with $r^2 \geq 0.8$ using HaploReg v4.2 and Roadmap data [13,14]. Annotated variants showed increased H3K27ac enhancer activity in astrocytes (rs35060225, rs34241561) and brain tissues including anterior caudate (rs10210927, rs12475444), hippocampus (rs12475444) and substantia nigra (rs12475444).

We have added the following information to the methods section (page 6, lines 207-219, 220-228, 229-232; page 7, lines 233-236):

In Silico analyses

“Shared genetic architecture between fatigue phenotypes in the cancer population studied and EHR-derived malaise and fatigue and chronic fatigue syndrome (ME/CFS) in white British participants of the UK Biobank was investigated using publicly available GWAS data [34,35]. Local genetic correlation r_g was calculated for susceptibility loci using the 1000 Genomes Project (EUR) Phase 3 as LD reference. Non-associated loci were filtered using univariate analysis and a Bonferroni-corrected significance level threshold, with bivariate tests performed for the remaining loci. The analysis was conducted using the package LAVA [36]. Colocalisation was explored using multi-tissue eQTL data derived from brain (hippocampus) and skeletal muscle tissue and whole blood (Genotype-Tissue Expression (GTEx) project v7) [37]. The analysis was conducted using the R package eQTLPlot [38]. Functional annotation was performed for regions

harbouring genome-wide association signals, candidate causal variants, and variants with $r^2 \geq 0.8$ using HaploReg v4.2 and Roadmap data [39,40].

Gene-based associations were assessed using MAGMA (v1.10) with GWAS summary statistics [41]. Genes were defined according to their genomic coordinates, including 35 kb upstream and 10 kb downstream regions to capture potential regulatory variants. SNP-level associations within each gene were aggregated using the mean SNP association model, which computes the average of SNP Z-scores while accounting for linkage disequilibrium through the SNP covariance matrix. LD was estimated using a reference panel from European ancestry individuals from the 1000 Genomes Project Phase 3. In the analysis, genes with a false discovery rate (FDR)-adjusted $P < 0.05$ were considered statistically significant. The R package qqman was used for creating the Manhattan plot.

Functional enrichment analyses were conducted using two gene sets: i) candidate genes mapped from the locus in proximity to the top GWAS-associated SNP as annotated by dbSNP and ii) genes identified through MAGMA analysis with nominal significance (unadjusted $P \leq 0.05$). Over-representation testing was conducted using clusterProfiler, ReactomePA, and related R packages across multiple databases, including Gene Ontology (biological process, cellular component, molecular function), KEGG, Reactome, and curated MSigDB collections (BIOCARTA, CGP, ImmuneSigDB, transcription factor and microRNA targets, WikiPathways) [42-47]. Enrichment results were exported and visualised with custom R scripts.”

The colocalisation analysis based on the GTEx cis-eQTLs was reported together with the results of the differential gene expression analysis using whole blood samples from the REQUITE cohort. The paragraph was moved up accordingly.

We have added the following information to the results section (page 9, lines 329-340; page 10, lines 341-355, 362-372):

“Local genetic correlations between incident physical fatigue and EHR-derived malaise and fatigue and ME/CFS were examined. Non-associated loci on chromosome 2 were filtered using univariate analysis and a Bonferroni-corrected significance level threshold of $\alpha = 0.05/220 = 0.00023$. Bivariate tests were performed for the remaining loci. For fatigue and malaise (3,413 cases, 403,789 controls), bivariate tests were calculated for 158 loci on chromosome 2 with a mean r_g of 0.03. 19 loci showed a Bonferroni-corrected significant bivariate r_g , including the locus around the index SNP rs142212041 ($r_g = -0.09$, $P = 0.0001$). For ME/CFS

(590 cases, 403,791 controls), bivariate tests were calculated for 158 loci, 18 of which showed a Bonferroni-corrected significant bivariate local genetic correlation. Across loci on chromosome 2, mean r_g was 0.03. For the locus harbouring the index SNP, a highly statistically significant local genetic correlation between incident physical fatigue and ME/CFS was observed ($r_g=0.18$, $P=5.79 \times 10^{-13}$). In the locus 200 kb around *ACTR3*, the P -values of the genome-wide association analysis and cis-eQTLs correlated most strongly for skeletal muscle tissue ($r=0.79$, $P=0.02$; Supplementary figure F) and whole blood ($r=0.63$, $P=0.10$; Supplementary figure G). No statistically significant associations were observed for pituitary gland ($r=0.47$, $P=0.21$; Supplementary figure H) and hippocampal tissue ($r=-0.17$, $P=0.89$; Supplementary figures I). Across tissues, no evidence was found that the trait-significant variant rs142212041 was associated with *ACTR3* expression levels. Differences in gene expression based on SNP-based genotypes were further explored in a different subset of the REQUITE PCa population. eQTL analysis was applied to 12 genes of 27 genes spanning the risk locus from 114.20 Mb to 115.00 Mb on chromosome 2, after excluding those with zero counts. After hard-calling the genotypes, 70 (58%) of 121 PCa patients with available dosage and expression data were included in the analysis. 60 patients (86%) were of homozygous (A/A) and 10 patients (14%) of heterozygous (A/B) genotype. Differential expression patterns were observed for the genes *CBWD2* and *ACTR3* with log2 fold changes of -0.17 and -0.09 ($P<0.04$), respectively, which were not statistically significant after correction for multiple testing (Supplementary table VI; Figure 4). (...) SNP annotation was performed for the index SNP rs142212041, candidate causal variants rs35060225, rs34241561, rs12472252, and variants with $r^2 \geq 0.8$. Annotated variants showed increased H3K27ac enhancer activity in astrocytes (rs35060225, rs34241561) and brain tissues including anterior caudate (rs10210927, rs12475444), hippocampus (rs12475444) and substantia nigra (rs12475444). In the gene-based association analysis, no genes reached statistical significance with an FDR-adjusted $P<0.05$. The most significant genes (unadjusted $P<0.001$) are shown in supplementary figure J. Functional enrichment of putative genes mapped from top GWAS SNPs suggested involvement of processes related to inflammation and cytoskeletal dynamics, including lamellipodium assembly/organization (Supplementary figure K and Supplementary table VII)."

We have added the following information regarding the Enrichment analysis to the discussion section (page 19, lines 445-450; page 20, lines 488-490):

“The gene-based analysis did not reveal significant associations; however, functional enrichment analysis of putative genes mapped from top GWAS SNPs suggested Gene Ontology processes related to lamellipodium assembly/organization. Lamellipodia are flat, actin-driven membrane protrusions that guide cell movement and support processes such as satellite cell migration during muscle repair [68]. Their dynamics may also be relevant in fatigue states. (...) Although the enrichment analysis suggested plausible pathways potentially involved in fatigue, the results are strongly influenced by *ACTR3*.”

We have added the following figures to the Appendix, which visualise the results of the colocalisation analyses (page 54-57), gene-based association analysis (page 59), and functional enrichment analysis of candidate genes (page 60):

eQTL analysis for Incident Physical Fatigue and ACTR3 In Muscle - Skeletal

Supplementary figure F. eQTL-GWAS colocalisation plot including eQTL enrichment and P-P correlation for the locus with the genome-wide statistically significant SNP rs142212041 and 200kb each around the candidate gene *ACTR3* based on the genome-wide association analysis for incident physical fatigue and skeletal muscle tissue eQTLs (GTEx v7).

eQTpLot analysis for Incident Physical Fatigue and ACTR3 In Whole Blood

Supplementary figure G. eQTL-GWAS colocalisation plot including eQTL enrichment and P-P correlation for the locus with the genome-wide statistically significant SNP rs142212041 and 200kb each around the candidate gene *ACTR3* based on the genome-wide association analysis for incident physical fatigue and whole blood eQTLs (GTEx v7).

eQTLPlot analysis for Incident Physical Fatigue and ACTR3 In Pituitary

Supplementary figure H. eQTL-GWAS colocalisation plot including eQTL enrichment and P-P correlation for the locus with the genome-wide statistically significant SNP rs142212041 and 200kb each around the candidate gene *ACTR3* based on the genome-wide association analysis for incident physical fatigue and pituitary gland eQTLs (GTEx v7).

eQTPlot analysis for Incident Physical Fatigue and ACTR3 In Brain - Hippocampus

Supplementary figure I. eQTL-GWAS colocalisation plot including eQTL enrichment and P-P correlation for the locus with the genome-wide statistically significant SNP rs142212041 and 200kb each around the candidate gene *ACTR3* based on the genome-wide association analysis for incident physical fatigue and hippocampal tissue eQTLs (GTEx v7).

Supplementary figure J. Manhattan plot for the gene-based association analysis (unadjusted $P < 0.001$) using the summary statistics of the genome-wide association analysis on incident physical fatigue up to two years following external beam radiotherapy in prostate cancer patients of the REQUITE cohort without fatigue before the start of radiotherapy.

Supplementary figure K. Functional enrichment analysis of candidate genes mapped from the locus in proximity to the top GWAS-associated SNP rs142212041 on chromosome 2. (A) Over-representation testing across multiple databases. (B) Biological Process Gene Ontology enrichment.

3. *“Methodology clarification:*

a. Provide rationale for genotype imputation thresholds (e.g., INFO ≥ 0.3), MAF cut-offs, and missingness filters.

b. Describe how missing covariate or phenotype data were handled (e.g., exclusion, imputation).

c. Justify the selection of covariates by including a table that lists all variables considered and the final model covariates.”

Covariates were selected in consultation with an interdisciplinary group of clinicians, researchers in the field of radiogenomics, and patient representatives. Available demographic, disease, and treatment data were selected based on their clinical relevance, whilst ensuring a sparse modelling approach to avoid overfitting. As REQUITE is a comprehensive data source, not all available potential covariates are listed. The case report form for the collection of patient factors at baseline before the start of radiotherapy, as described in Seibold et al., is included in Appendix A of this document [15].

We have added the following information to the methods section (page 4, lines 151-152; see also Reviewer 1, comment 5):

“To facilitate comparability of findings with previous studies on toxicities after radiotherapy, the analysis was based on common SNPs with MAF ≥ 0.05 and imputation score ≥ 0.3 [22-24].”

Explanations on the handling of missing phenotype data are provided on page 4, line 160; page 5, lines 161-162:

“In both questionnaires, missing items were replaced by mean values of the completed items if at least half of the items of the scale were provided by a participant [225].”

The sequential approach to minimise missing T stage data is described on page 5, lines 190-191:

“(…) If available, clinical T stage was considered, else pathological T stage or MRI T stage.), (…)”

We have added the following information to the methods section (page 5, line 193):

“Patients with complete data were included in the regression models.”

4. *“Expand limitations:*

a. Explicitly acknowledge limited power to detect small-effect variants, wide CIs, and the preliminary nature of findings without replication.

b. Discuss the subjectivity of fatigue assessments, implications of dichotomization, and potential misclassification.

c. Note analyses restricted to primarily European ancestry limit applicability to other populations.

d. Acknowledge potential influences of comorbidities, lifestyle factors, and psychosocial variables not fully captured.

e. Discuss that whole-blood eQTL may not reflect relevant tissues (CNS, muscle) and that in silico eQTL from GTEx have limitations (post-mortem tissue, older donors).”

We have expanded the limitations as suggest and added the following information to the discussion section (page 18, lines 433-434; page 19, lines 472-477; page 20, lines 478-488, 499-501):

“However, GTEx samples were collected from deceased subjects, which may affect expression data [65]. (...) The presented work has some limitations. The dichotomisation of fatigue assessments and tThe sample size, modest for a genome-wide analysis, limited the statistical power to identify genetic variants with smaller effect sizes. Furthermore, low-frequency and rare variants with MAF <0.05 were not included in this analysis. Although the genome-wide statistically significant SNP for incident physical fatigue was observed in the subgroup of men unfatigued prior to radiotherapy, this finding could be biased by non-radiation related manifestation of fatigue. The available data do not allow to establish radiotherapy as causative for the observed fatigue, limiting the interpretation of the findings. However, the genome-wide statistically significant SNP for physical fatigue was observed in the subgroup of men without fatigue prior to radiotherapy after adjustment for demographic, disease- and treatment-specific characteristics. Therefore, it might be assumed that the fatigue that developed during radiotherapy is largely related to the treatment, although other factors such as anxiety, stress, pain, or insomnia due to the cancer diagnosis may have contributed. As no validated threshold for clinically important fatigue was previously published for the MFI, patients were classified as fatigued and non-fatigued using population-based reference values, as opposed to sample-based cut-offs. Measurement error as well as comorbidities and lifestyle and psychosocial factors not fully captured may lead to residual confounding. (...) It is still unclear whether the findings can be generalised to patients with metastatic disease at diagnosis, other tumour entities, or populations of non-European ancestry.”

5. *“Data and code availability:*

a. *Commit to publicly sharing de-identified summary statistics (e.g., via GWAS Catalog or institutional repository) to enable future meta-analyses.*

b. *Make analysis pipelines (QC, association tests, resampling scripts, phenotype derivation) available (e.g., GitHub) to enhance reproducibility.”*

We aim to submit the summary statistics of this analysis to *The NHGRI-EBI Catalogue of human genome-wide association studies*. Phenotype derivation is reproducible using the EORTC QLQ-C30 scoring manual and MFI publications [16,17]. The pipeline for the association analysis is available on request from the corresponding author.

6. *“Interpretation and discussion:*

a. *Reframe statements about the lead association as “hypothesis-generating” rather than definitive. For example: “We identify a locus near ACTR3/CBWD2 that merits further investigation; replication is required to confirm this signal.”*

b. *Avoid overemphasis on biological interpretation before in silico and functional corroboration; instead, present the mechanistic links as plausible but unconfirmed.*

c. *Highlight how in silico annotations (eQTL, regulatory overlap), pathway enrichment, and phenotype correlations (PRS/genetic correlation) together suggest biological plausibility, even if each line of evidence alone is insufficient.*

d. *Discuss how these convergent data justify prioritizing this locus for future functional studies.*

e. *Explain what SNP-based heritability estimates (e.g., $H^2 \approx 0.31 \pm 0.18$ for overall fatigue) imply about genetic contribution vs. environmental/noise, comparing with prior heritability estimates for fatigue or related traits.*

f. *Discuss limitations of heritability estimation (e.g., measurement error, sample size).*

g. *Outline concrete next steps: forming collaborations for meta-analysis, collecting similar phenotype data in other cohorts, functional studies (in vitro/in vivo), and exploring intervention targets if findings hold.*

h. *Emphasize that while replication is ideal, the current analyses (robustness checks, in silico triangulation) build a foundation for prioritizing loci and guiding experimental validation.”*

We have reviewed the manuscript to avoid strong mechanistic interpretations. To highlight that the results presented are initial evidence requiring further replication, we have made the following additions to the discussion section (page 20, lines 501-502; page 20, lines 509-510):

“The identified susceptibility locus near *ACTR3/CBWD2* points to a plausible mechanism, which remains unconfirmed without functional corroboration. (...) In conclusion, this study emphasizes the multi-dimensionality of fatigue and has provided initial evidence for its heritability across different fatigue phenotypes.”

The complementary in silico analyses were addressed in the following section of the discussion (page 18, lines 428-441; page 19, lines 442-443):

Consistently, a highly statistically significant local genetic correlation between incident physical fatigue and ME/CFS was observed for the locus harbouring the index SNP rs142212041, albeit of weak magnitude. Functional studies could prioritise this locus to investigate shared mechanisms underlying the aetiology of both phenotypes. GWAS and GTEx cis-eQTL signals correlated strongly in skeletal muscle tissue, although the trait-significant variant was not associated with *ACTR3* expression levels. However, GTEx samples were collected from deceased subjects, which may affect expression data [65]. In patients of the REQUITE cohort treated with radiotherapy, differences in the expression of the *ACTR3* and *CBWD2* genes were observed between SNP-based genotypes in whole blood, although not statistically significant after correction for multiple testing. Due to the hard-calling of imputed dosages, only a subset of samples (58%) with expression data were included. This might have limited the detection of subtle expression differences. Yet, the results may inform further research on fatigue and the genes in the identified risk locus, which is in proximity to a region between the *PAX8* and *CBWD2* genes, which has been previously shown to be linked to usual sleep duration [4666]. Consistent with recent findings, functional analysis of regulatory elements indicated increased enhancer activity in astrocytes and several brain regions [67].

Heritability estimates and their limitations have been amended to the discussion (page 19, lines 443-445, 451-455):

~~The estimated heritability for fatigue suggests that identification of further susceptibility alleles could require additional large studies.~~ (...) Heritability estimates suggest that genetic factors may contribute to fatigue in PCa patients. The proportion of phenotypic variation explained by common

SNPs varied between fatigue dimensions and was generally higher than estimated for non-cancer populations [69]. Large standard errors, which may be inflated due to the small sample size, limit the conclusions that can be drawn and emphasise the need for collecting fatigue assessments in larger cohorts.

References

- [1] Kerns SL, Dorling L, Fachal L, Bentzen S, Pharoah PD, Barnes DR, Gómez-Caamaño A, Carballo AM, Dearnaley DP, Peleteiro P, Gulliford SL, Hall E, Michailidou K, Carracedo Á, Sia M, Stock R, Stone NN, Sydes MR, Tyrer JP, Ahmed S, Parliament M, Ostrer H, Rosenstein BS, Vega A, Burnet NG, Dunning AM, Barnett GC, West CM; Radiogenomics Consortium. Meta-analysis of Genome Wide Association Studies Identifies Genetic Markers of Late Toxicity Following Radiotherapy for Prostate Cancer. *EBioMedicine*. 2016 Aug;10:150-63. doi: 10.1016/j.ebiom.2016.07.022. Epub 2016 Jul 20. PMID: 27515689; PMCID: PMC5036513.
- [2] Farazi M, Yang X, Gehl CJ, Barnett GC, Burnet NG, Chang-Claude J, Parker CC, Dunning AM, Azria D, Choudhury A, Rancati T, De Ruyscher D, Seibold P, Sperk E, Talbot CJ, Veldeman L, Webb AJ, Elliott R, Aguado-Barrera ME, Carballo AM, Fuentes-Ríos O, Gómez-Caamaño A, Peleteiro P, Vega A, Ostrer H, Rosenstein BS, Saito S, Parliament M, Usmani N, Marples B, Chen Y, Morrow G, Messing E, Janelins MC, Hall W, West CML, Auer PL, Kerns SL. A Polygenic Risk Score for Late Bladder Toxicity Following Radiotherapy for Non-Metastatic Prostate Cancer. *Cancer Epidemiol Biomarkers Prev*. 2025 May 2;34(5):795-804. doi: 10.1158/1055-9965.EPI-24-1228. PMID: 40029246; PMCID: PMC12048210.
- [3] Jandu HK, Veal CD, Fachal L, Luccarini C, Aguado-Barrera ME, Altabas M, Azria D, Baten A, Bourgier C, BghBultijnck R, Colciago RR, Farcy-Jacquet MP, Chang-Claude J, Choudhury A, Dunning A, Elliott RM, Green S, Gutiérrez-Enríquez S, Herskind C, Lambrecht M, Monten C, Rancati T, Reyes V, Rosenstein BS, De Ruyscher D, Carmen De Santis M, Seibold P, Sperk E, Veldwijk M, Paul Symonds R, Stobart H, Taboada-Valladares B, Vega A, Veldeman L, Webb AJ, Weltens C, West CM, Rattay T, Talbot CJ; REQUITE consortium. Genome-wide association study of treatment-related toxicity two years following radiotherapy for breast cancer. *Radiother Oncol*. 2023 Oct;187:109806. doi: 10.1016/j.radonc.2023.109806. Epub 2023 Jul 10. PMID: 37437607.
- [4] Forde A, Hemani G, Ferguson J. Review and further developments in statistical corrections for Winner's Curse in genetic association studies. *PLoS Genet*. 2023 Sep 18;19(9):e1010546. doi: 10.1371/journal.pgen.1010546. PMID: 37721937; PMCID: PMC10538662.
- [5] Ghosh A, Zou F, Wright FA. Estimating odds ratios in genome scans: an approximate conditional likelihood approach. *Am J Hum Genet*. 2008 May;82(5):1064-74. doi: 10.1016/j.ajhg.2008.03.002. Epub 2008 Apr 24. Erratum in: *Am J Hum Genet*. 2008 May;82(5):1224. PMID: 18423522; PMCID: PMC2665019.
- [6] Zou Y, Carbonetto P, Wang G, Stephens M. Fine-mapping from summary data with the "Sum of Single Effects" model. *PLoS Genet*. 2022 Jul 19;18(7):e1010299. doi: 10.1371/journal.pgen.1010299. PMID: 35853082; PMCID: PMC9337707.
- [7] 798: Malaise and fatigue [Internet]. [accessed 2025 Jul 18]. Available from: <https://pheweb.org/UKB-TOPMed/pheno/798>
- [8] 798.1: Chronic fatigue syndrome [Internet]. [accessed 2025 Jul 18]. Available from: <https://pheweb.org/UKB-TOPMed/pheno/798.1>
- [9] Werme J, van der Sluis S, Posthuma D, de Leeuw CA. An integrated framework for local genetic correlation analysis. *Nat Genet*. 2022 Mar;54(3):274-282. doi: 10.1038/s41588-022-01017-y. Epub 2022 Mar 14. PMID: 35288712.
- [10] GTEx Consortium. The Genotype-Tissue Expression (GTEx) project. *Nat Genet*. 2013 Jun;45(6):580-5. doi: 10.1038/ng.2653. PMID: 23715323; PMCID: PMC4010069.
- [11] Drivas TG, Lucas A, Ritchie MD. eQTpLot: a user-friendly R package for the visualization of colocalization between eQTL and GWAS signals. *BioData Min*. 2021 Jul 17;14(1):32. doi: 10.1186/s13040-021-00267-6. PMID: 34273980; PMCID: PMC8285863.
- [12] Stanfill AG, Cao X. Enhancing Research Through the Use of the Genotype-Tissue Expression (GTEx) Database. *Biol Res Nurs*. 2021 Jul;23(3):533-540. doi: 10.1177/1099800421994186. Epub 2021 Feb 18. PMID: 33596660; PMCID: PMC8191158.
- [13] Ward LD, Kellis M. HaploReg v4: systematic mining of putative causal variants, cell types,

regulators and target genes for human complex traits and disease. *Nucleic Acids Res.* 2016 Jan 4;44(D1):D877-81. doi: 10.1093/nar/gkv1340. Epub 2015 Dec 10. PMID: 26657631; PMCID: PMC4702929.

[14] Bernstein BE, Stamatoyannopoulos JA, Costello JF, Ren B, Milosavljevic A, Meissner A, Kellis M, Marra MA, Beaudet AL, Ecker JR, Farnham PJ, Hirst M, Lander ES, Mikkelsen TS, Thomson JA. The NIH Roadmap Epigenomics Mapping Consortium. *Nat Biotechnol.* 2010 Oct;28(10):1045-8. doi: 10.1038/nbt1010-1045. PMID: 20944595; PMCID: PMC3607281.

[15] Seibold P, Webb A, Aguado-Barrera ME, Azria D, Bourcier C, Brengues M, Briers E, Bultijnck R, Calvo-Crespo P, Carballo A, Choudhury A, Cicchetti A, Claßen J, Delmastro E, Dunning AM, Elliott RM, Fachal L, Farcy-Jacquet MP, Gabriele P, Garibaldi E, Gómez-Caamaño A, Gutiérrez-Enríquez S, Higginson DS, Johnson K, Lobato-Busto R, Mollà M, Müller A, Payne D, Peleteiro P, Post G, Rancati T, Rattay T, Reyes V, Rosenstein BS, De Ruyscher D, De Santis

MC, Schäfer J, Schnabel T, Sperk E, Symonds RP, Stobart H, Taboada-Valladares B, Talbot CJ, Valdagni R, Vega A, Veldeman L, Ward T, Weißenberger C, West CML, Chang-Claude J; REQUITE consortium. REQUITE: A prospective multicentre cohort study of patients undergoing radiotherapy for breast, lung or prostate cancer. *Radiother Oncol.* 2019 Sep;138:59-67. doi: 10.1016/j.radonc.2019.04.034. Epub 2019 May 27. PMID: 31146072.

[16] Fayers PM, Aaronson NK, Bjordal K, Groenvold M, Curran D, Bottomley A, on behalf of the EORTC Quality of Life Group. The EORTC QLQ-C30 Scoring Manual (3rd Edition). European Organisation for Research and Treatment of Cancer, Brussels 2001.

[117] Smets EM, Garssen B, Bonke B, De Haes JC. The Multidimensional Fatigue Inventory (MFI) psychometric qualities of an instrument to assess fatigue. *J Psychosom Res.* 1995 Apr;39(3):315-25. doi: 10.1016/0022-3999(94)00125-o. PMID: 7636775.

Additional edits

We have changed the order of the co-authors:

- a. Miquel E. Aguado Barrera is now shared second author as he substantially contributed to the additional comprehensive pathway analysis.
- b. Erik Briers and Renée Bultijnck have been exchanged in the order of co-authors to maintain the alphabetical order.

Affiliations and funding statements have been adjusted.

- The following phrasing was changed in the abstract (page 2, lines 73-75):

“A total of 1,358 men recorded fatigue using the EORTC QLQ-C30 and 869 men additionally completed the Multidimensional Fatigue Inventory (MFI) up to two years post-radiotherapy.”

- We have written out the number of follow-up years.

- Page 4, line 126:

“(…) before radiotherapy initiation, at the end of radiotherapy, and annually thereafter until at least 2two years later [15].”

- Page 5, line 163:

“(…) proposed threshold for clinically important levels as scores ≥ 39 at 4one and/or 2two years (…)”

- Page 5, lines 165-166:

“(…) available for the MFI, prevalent fatigue was defined as scores ≥ 75 th percentile at 4one and/or 2two years after the end of radiotherapy (…)”

- A comparative sign has been corrected (page 5, line 165):

“(…) prevalent fatigue was defined as scores ≥ 75 th percentile at one and/or two years after the end of radiotherapy (…)”

- Adjustments to the manuscript required the following changes to abbreviations and the labelling of tables and figures:

- Page 2, line 75:

“(...) completed the Multidimensional Fatigue Inventory (MFI) up to two years post-radiotherapy.”

- Page 10, line 354-355:

“(...) which were not statistically significant after correction for multiple testing (Supplementary table IV_I; Figure 4).”

- Page 18, lines 416-418:

“(...) the *ACTR3* gene was shown to be differentially expressed in women with major depressive disorder and patients with ~~myalgic encephalomyelitis/chronic fatigue syndrome (ME/CFS)~~ (...)”

- Page 58:

“Supplementary table V_I. Results of the differential gene expression analysis by SNP-based genotypes for incident long-term physical fatigue in the REQUITE cohort (separate from the GWAS sample).”

- The following modifications have been made to the wording:
 - Page 20, line 509:

“In conclusion, this study ~~emphasizes~~ emphasises the multi-dimensionality of fatigue and has provided initial evidence for its heritability across different fatigue phenotypes.”

- Missing hyphens have been added:
 - Page 8, line 275:

“(...) in the intermediate- or high-risk groups for biochemical recurrence (...)”

- We have changed the reporting of *P*-values from P_{\square} to P_{\square} in the manuscript.
- We have added page numbers to the manuscript.

Appendix A

Case report form for the collection of patient factors at baseline before the start of radiotherapy for prostate cancer patients in the REQUITE cohort.

P2a

PROSTATE PATIENT FACTORS – BASELINE (to be completed pre-radiotherapy)

Study Number RQ□□□□□-□

Patient Initials □□□□

Date of Birth (dd/mm/yyyy) □□/□□/□□□□

Date Completed (dd/mm/yyyy) □□/□□/□□□□

Name + Signature of Person completing the CRF _____

Patient Information

Height (cm) □□□ **Weight (kg)** □□□

Age at start of radiotherapy (yrs) □□

Smoker 0=Never
 1=Ex before cancer diagnosis
 2=Ex since cancer diagnosis
 3=Current
 7=Do not wish to answer

If ever smoker **Duration of smoking (yrs)** □□

No. of tobacco products (e.g. cigarettes) a day □□

If ex smoker before cancer diagnosis:
Time since quitting smoking (yrs)

Tobacco product _____

Alcohol intake 0=Never
 1=Previously consumed alcohol, but stopped BEFORE cancer diagnosis
 2=Previously consumed alcohol, but stopped AT cancer diagnosis
 3=Current
 7=Do not wish to answer

Previous alcohol consumption: **Approximate number of alcoholic drinks a week**
 777=Do not wish to answer
 888=Not applicable

Current alcohol consumption: **Approximate number of alcoholic drinks a week**
 777=Do not wish to answer
 888=Not applicable

Diabetes 0=No 1=Yes **If yes, duration (yrs)** □□

Rheumatoid Arthritis 0=No 1=Yes **If yes, duration (yrs)** □□

RQ□□□□□-□

Systemic Lupus Erythematosus	[ ]	0=No 1=Yes	If yes, duration (yrs)	[ ] [ ]
Other collagen vascular disease	[ ]	0=No 1=Yes	If yes, duration (yrs)	[ ] [ ]
Hypertension	[ ]	0=No 1=Yes	If yes, duration (yrs)	[ ] [ ]
History of heart disease	[ ]	0=No 1=Yes	If yes, duration (yrs)	[ ] [ ]
Any inflammatory bowel or diverticular disease	[ ]	0=No 1=Crohn's disease 2=Colitis ulcerosa 3=Diverticulosis 4=Other	If yes, duration (yrs)	[ ] [ ]
Haemorrhoids	[ ]	0=No 1=Yes	If yes, duration (yrs)	[ ] [ ]
			If yes, physician confirmed?	[ ] 0=No 1=Yes 9=Not known
Depression	[ ]	0=No 1=Yes	If yes, duration (yrs)	[ ] [ ]
Medication at cancer diagnosis				
On ACE inhibitor?	[ ]	0=No 1=Yes	If yes, duration (yrs)	[ ] [ ]
On a beta blocker?	[ ]	0=No 1=Yes	If yes, duration (yrs)	[ ] [ ]
On other anti-hypertensive drug?	[ ]	0=No 1=Yes	If yes, duration (yrs)	[ ] [ ]
On statin?	[ ]	0=No 1=Yes	If yes, duration (yrs)	[ ] [ ]
On other lipid-lowering drugs?	[ ]	0=No 1=Yes	If yes, duration (yrs)	[ ] [ ]
On anti-diabetic drug?	[ ]	0=No 1=Yes	If yes, duration (yrs)	[ ] [ ]
On phosphodiesterase type 5 (PDE5) inhibitor like cialis?	[ ]	0=No 1=Yes	If yes, duration (yrs)	[ ] [ ]
On sildenafil?	[ ]	0=No 1=Yes	If yes, duration (yrs)	[ ] [ ]
On 5 alpha-reductase inhibitor?	[ ]	0=No 1=Yes	If yes, duration (yrs)	[ ] [ ]
On alpha blocker?	[ ]	0=No 1=Yes	If yes, duration (yrs)	[ ] [ ]
On anti-muscarinic drug?	[ ]	0=No 1=Yes	If yes, duration (yrs)	[ ] [ ]
On amiodarone?	[ ]	0=No 1=Yes	If yes, duration (yrs)	[ ] [ ]

RQ□□□□□-□

On analgesics? 0=No
1=Yes

On antidepressant? 0=No
1=Yes

Hip replacement? 0=No
1=Unilateral
2=Bilateral

Bladder TUR 0=No
1=Yes

Family history of prostate cancer in first degree relative 0=No
1=Yes

If yes, duration (yrs)

If yes, duration (yrs)

Previous abdominal surgery 0=No
1=Appendectomy
2=Cholecystectomy
3=Rectum-sigma resection
4=Nephrectomy
5=Other

Family history of radiotherapy toxicity 0=No
1=Yes
9=Not known

Other co-morbidity _____

Previous Malignancies? 0=No
1=Yes

Which type?
ICD-10 / ICD-O-3 coding: /

Date of diagnosis (dd/mm/yyyy) / /

Therapy received for previous malignancy

Surgery 0=No
1=Yes

Hormonal therapy 0=No
1=Yes

Chemo therapy 0=No
1=Yes

Radio therapy 0=No
1=Yes

Other therapy 0=No
1=Yes

No therapy 0=No
1=Yes

Date of last therapy for previous malignancy (dd/mm/yyyy) / /

Ethnicity

1=White (European or American European)
2=White and Black Caribbean Mixed
3=White and Black African Mixed
4=White and Asian Mixed
5=Hispanic American
6=Turkish
7=Indian
8=Pakistani
9=Bangladeshi
10=Chinese
11=Japanese

12=Other Asian
13=Black Caribbean
14=Black African
15=Northern African
16=African American
17=Jewish Ashkenazi
18=Jewish Sephardi
19=Any Other Ethnic Background; please specify _____
77=Patient refused to give answer

Highest educational/professional qualification received?

- 1=Primary school
- 2=Secondary school (Please select an option _____)
- 3=Professional school (e.g. technical. Please specify type _____)
- 4=University (or equivalent)
- 5=Others, please specify _____
- 7=Do not wish to answer

- Options for "Secondary school":
- a. UK: GSCCE / O level
 - b. UK: A level
 - c. US: High school
 - d. B: Algemeen Secundair Onderwijs
 - e. GER: Hauptschule
 - f. GER: Realschule/Mittlere Reife
 - g. GER: Gymnasium/Abitur
 - h. CH: Realschule
 - i. CH: Sekundarschule
 - j. CH: Gymnasium / Matura
 - k. F: college
 - l. F: lycée/baccalaureate
 - m. I: scuola secondaria di primo grado
 - n. I: scuola secondaria di secondo grado
 - o. NL: voortgezet onderwijs
 - p. SP: Educación Secundaria Obligatoria/Bachillerato
 - q. Other, please specify _____

Net household income (average) per month

- 1=<1.000 €
- 2=1.000-<2.000€
- 3=2.000-<3.000€
- 4=3.000-<4.000€
- 5=4.000-<5.000€
- 6=5.000-<6.000€
- 7=6.000-<7.000€
- 8=7.000-<8.000€
- 9=8.000€ and higher
- 77= Do not wish to answer

Number of household members

RESPONSE TO REVIEWERS' COMMENTS

All changes in the manuscript were added in editing mode for easy identification. We sincerely thank the reviewer for their constructive feedback. Below you will find a point-by-point list of the changes we have made:

Reviewer 1

1. *“A description and modeling of missingness (including whether baseline fatigue predicts loss)”*

To describe the availability and missingness of fatigue assessments in the EORTC QLQ-C30 and Multidimensional Fatigue Inventory (MFI) questionnaires up to two years after radiotherapy in prostate cancer patients of the REQUITE cohort, the following figure has been added to the supplementary materials (page 30) and referenced in the results section (page 7, lines 264-265):

Supplementary figure A. Flow diagram on the availability of long-term fatigue assessments using the EORTC QLQ-C30 and Multidimensional Fatigue Inventory up to two years after radiotherapy in prostate cancer patients of the REQUITE cohort.

To assess the prognostic value of fatigue at baseline prior to radiotherapy for missing fatigue scores one and/or two years later, logistic regression models were calculated for the fatigue scale in the EORTC QLQ-C30 and for each fatigue dimension in the MFI. Clinically important levels of physical fatigue (for which the genome-wide significant SNP was observed) before the start of radiotherapy were not statistically significantly ($P=0.78$) associated with missing fatigue assessments one and/or two years later in 972 prostate cancer patients receiving external beam radiotherapy without brachytherapy. No statistically significant association ($P=0.51$) was observed after restricting the analysis to 940 patients with complete, non-imputed physical fatigue assessments at baseline before radiotherapy initiation. Furthermore, no statistically significant associations were found between baseline fatigue levels and missing long-term fatigue scores for the EORTC QLQ-C30 fatigue scale and for the mental fatigue, reduced activity, and reduced motivation dimensions in the MFI. For general fatigue in the MFI, pre-radiotherapy fatigue level predicted missingness of fatigue assessments one and/or two years later in 971 prostate cancer patients (OR: 1.67, 95% CI: 1.01-2.66, $P=0.04$). The association remained significant after adjustment for age.

The results of the regressions models do not suggest outcome-dependent loss to follow-up up to two years after radiotherapy in the sample of prostate cancer patients studied, except for general fatigue in the MFI.

2. *“A primary analysis that avoids mean imputation (e. g. mixed - effects maximum likelihood or multiple imputation)”*

Mean imputation as recommended in the scoring manual for the EORTC QLQ-C30 was performed for both questionnaires [1]. Only if two of the three items comprising the fatigue scale in the EORTC QLQ-C30 were completed by a patient, the missing third item was replaced by the mean value of the item ratings provided by the patient. Consistently, missing items in the MFI were only replaced by the mean value if at least half of the four items for a particular fatigue dimension had been completed by a patient.

The vast majority ($\geq 95.1\%$) of fatigue surveys collected one and/or two years after radiotherapy were fully available. The proportion of long-term fatigue scores in which mean-substituted items were considered was 2.5% (35/1,381) for the EORTC QLQ-C30 fatigue scale, 4% (35/877) for general fatigue, 2.5% (22/876) for physical fatigue, 4.9% (43/873) for mental fatigue, 3.4% (30/874) for reduced activity, and 2.7% (24/874) for reduced motivation in the MFI. The

proportions of fatigue scores containing mean-substituted item(s) have been added to the manuscript in Supplementary figure A (page 30).

Mean substitution can be considered the standard procedure for dealing with missing values in the fatigue measurement instruments used and was applied to only a small proportion of patients who assessed at least half of item on the given scale/dimension. In order to ensure the comparability of our results with previous and future findings, we suggest retaining the mean-substituted fatigue assessments in the primary analysis.

3. *“Sensitivity analyses using inverse probability weighting (IPW) and MNAR (delta-adjusted) approaches for the lead SNPs.”*

We acknowledge that the complete case re-analysis performed does not constitute a valid test of the mechanisms underlying the missing long-term fatigue data. Therefore, potential bias due to outcome-dependent dropout was investigated using regression models assessing the prognostic value of baseline pre-radiotherapy fatigue levels for the missingness of fatigue scores one and/or two years later (see comment 1).

Several statistical methods have been proposed for imputing incomplete ordinal variables, including the use of an ordered probit model with sample selection and predictive mean matching [2,3,4]. Fatigue item ratings provided by prostate cancer patients are generally characterised by right-skewed, zero-inflated distributions, as the majority of patients score the lowest category of the Likert item corresponding to low fatigue levels (Figure R2). While predictive mean matching usually preserves the original distribution of the data better than parametric multiple imputation approaches, it has been shown that its robustness can be limited by severe skewness [5].

We agree that additional sensitivity analyses on MNAR could further substantiate the robustness of the findings in a confirmatory setting. Given the exploratory nature of the analysis presented, aiming to stimulate mechanistic studies on the potential susceptibility locus identified (page 20, lines: 497-504), and the extensive complementary sensitivity analyses performed during the first revision, we believe that additional model-based sensitivity analyses provide limited added value.

EORTC QLQ-C30 Fatigue Scale

Multidimensional Fatigue Inventory

Figure R2. Fatigue item assessments using the EORTC QLQ-C30 and Multidimensional Fatigue Inventory one year after radiotherapy in prostate cancer patients of the REQUITE cohort.

Additional edits

- We have adjusted the following information in the results section to clarify how the analysed sample was derived from available fatigue data in the EORTC QLQ-C30 and MFI one and/or two years after radiotherapy (page 7, lines 264-268; page 8, lines 269-270):

“Of ~~1,760~~1,578 PCa patients enrolled in the REQUITE cohort prior to radiotherapy, ~~1,578 (90%)~~ were treated with EBRT without brachytherapy (Supplementary figure A). Genetic data were available for 1,508 patients. Data on the ~~three-item~~ fatigue scale from the EORTC QLQ-C30 questionnaire were available for ~~1,358 (86%)~~ 1,381 (1,381/1,508; 92%) participants with available genetic data, with ~~869 (55%)~~ 877 (877/1,508; 58%) participants reporting at least one of the five fatigue dimensions in the MFI questionnaire ~~up to~~ at one and/or two years after the end of radiotherapy (Table 1).”

- To facilitate comparison with Supplementary figure A (page 30), the proportions of fatigued patients in the EORTC QLQ-C30 and MFI were reported separately from the availability of covariate data. Table 1 has been adjusted accordingly and the following adjustments have been made in the abstract (page 2, line 77) and results section (page 8, lines 278-282):

“A total of ~~1,358~~1,381 men recorded fatigue using the EORTC QLQ-C30 and ~~869~~877 men additionally completed the Multidimensional Fatigue Inventory (MFI) up to two years post-radiotherapy. (...) ~~21% (279) of 1,358~~ 20% (283) of 1,381 PCa patients who completed the EORTC QLQ-C30 reported being fatigued ~~up to~~ at one and/or two years following radiotherapy. ~~32% (274/868)~~ 31% (274/876) of men with MFI assessments experienced physical, ~~31% (269/869)~~ 31% (269/877) general, and ~~29% (255/865)~~ 29% (256/873) mental fatigue. 28% reported reduced activity (~~246/866~~)(247/874) and motivation (~~244/866~~)(246/874).”

- The addition of Supplementary figure A required the labelling of the remaining supplementary tables and figures in the manuscript and appendix to be adjusted.
- A correction to British English has been made (page 8, line 274):

“(…) localiszed disease (…)”

- An extra space and missing space were corrected on page 9, line 247 and page 10, line 322.
- The address of affiliation 23 has been changed:

“23 Centro de Investigación en Red de Enfermedades Raras (CIBERER), Santiago de Compostela, 1570628029, Madrid, Spain.”

Correction of a covariate the primary analysis

We corrected an error in the primary genome-wide association analysis caused by the erroneous omission of a covariate (baseline fatigue level). By including the omitted covariate, the number of patients included in the primary analyses was reduced from N=1,358 to 1,284 for the EORTC QLQ-C30 fatigue scale and from N=869 to N=826 for the MFI. As in the previously misspecified primary models, no SNP in the overall cohort of prostate cancer patients was genome-wide statistically significantly associated with long-term fatigue in the EORTC QLQ-C30 and/or MFI. Therefore, the conclusions of the primary analysis do not change. We have corrected the respective Manhattan (Figure 1) and QQ (Supplementary figure B) plots, the upper panel of the LocusZoom plot (Figure 3), and Supplementary table II accordingly.

The secondary analysis in which the genome-wide statistically significant SNP for incident physical fatigue was observed and all complementary sensitivity, subgroup, and in silico analyses are not affected, as they refer to the subgroup of men without fatigue at baseline (for whom **no adjustment for baseline fatigue level was performed**). Therefore, also the overall conclusions of the manuscript with the top hits stay the same.

We apologise for not noticing this earlier.

We have made the following adjustments in the results section (page 8, lines 293-300):

“The associated QQ plots did not indicate genomic inflation ($\lambda \leq 1.045$) and indicated more variants associated with fatigue phenotypes than expected by chance (Supplementary figure AB). (...) Associations with $P < 5 \times 10^{-6}$ were found for the index SNPs rs74800609 ($P = 3.3 \times 10^{-7}$), rs17710795 ($P = 4.5 \times 10^{-7}$), and rs35612476 ($P = 9.5 \times 10^{-7}$) for general fatigue, rs1480885 ($P = 9.003.7 \times 10^{-78}$), rs56188740 ($P = 1.00 \times 10^{-7}$), and rs12455002 ($P = 7.9 \times 10^{-7}$) for reduced activity, rs13394124 ($P = 6.60 \times 10^{-7}$) for physical fatigue, and rs35837098 ($P = 1.852 \times 10^{-7}$) for fatigue based on the EORTC QLQ-C30. The Manhattan plots for reduced activity further indicates several SNPs aligned with the index rs74800609 on chromosome 14 for general fatigue and index rs1480885 on chromosome 8 for reduced activity.

References

[1] Fayers PM, Aaronson NK, Bjordal K, Groenvold M, Curran D, Bottomley A, on behalf of the EORTC Quality of Life Group. The EORTC QLQ-C30 Scoring Manual (3rd Edition). European Organisation for Research and Treatment of Cancer, Brussels 2001.

[2] Little, R. J., Carpenter, J. R., Lee, K. J. A comparison of three popular methods for handling missing data: complete-case analysis, inverse probability weighting, and multiple imputation. *Sociological Methods & Research*. 2024; 53(3): 1105-1135.

[3] Hammon, A. Multiple imputation of ordinal missing not at random data. *ASta Advances in Statistical Analysis*. 2023; 107(4), 671-692.

[4] Austin PC, van Buuren S. Imputation of incomplete ordinal and nominal data by predictive mean matching. *Stat Methods Med Res*. 2025 Aug 17:9622802251362642. doi: 10.1177/09622802251362642. Epub ahead of print. PMID: 40820317.

[5] Kleinke, K. Multiple Imputation Under Violated Distributional Assumptions: A Systematic Evaluation of the Assumed Robustness of Predictive Mean Matching. *Journal of Educational and Behavioral Statistics*. 2017; 42(4), 371-404.